# *Drosophila* Evi5 is a critical regulator of intracellular iron transport via transferrin and ferritin interactions

Sattar Soltani[1], Samuel M. Webb[2], Thomas Kroll[2] & Kirst King-Jones [1] ✉

Vesicular transport is essential for delivering cargo to intracellular destinations. Evi5 is a Rab11-GTPase-activating protein involved in endosome recycling. In humans, Evi5 is a high-risk locus for multiple sclerosis, a debilitating disease that also presents with excess iron in the CNS. In insects, the prothoracic gland (PG) requires entry of extracellular iron to synthesize steroidogenic enzyme cofactors. The mechanism of peripheral iron uptake in insect cells remains controversial. We show that Evi5-depletion in the *Drosophila* PG affected vesicle morphology and density, blocked endosome recycling and impaired trafficking of transferrin-1, thus disrupting heme synthesis due to reduced cellular iron concentrations. We show that ferritin delivers iron to the PG as well, and interacts physically with Evi5. Further, ferritin-injection rescued developmental delays associated with Evi5-depletion. To summarize, our findings show that Evi5 is critical for intracellular iron trafficking via transferrin-1 and ferritin, and implicate altered iron homeostasis in the etiology of multiple sclerosis.

Abnormal iron levels are associated with a series of disorders, including iron deficiency anemia, iron overload hemochromatosis, porphyrias and neurodegenerative disorders such as multiple sclerosis (MS), Lou Gehrig's and Alzheimer's disease[1,2]. Biological iron is most commonly found in two types of protein cofactors, heme and iron-sulfur clusters (Fe-S), and act in various electron transfer reactions[3,4]. Less common are proteins that bind the metal directly, such as mononuclear (non-heme) iron oxygenases or dinuclear iron centers present is some ferroxidases[5]. Consequently, iron functions in a broad range of processes, including oxygen transport by hemoglobin, energy production by mitochondrial cytochromes, and in diverse enzymatic pathways such as DNA synthesis and steroid hormone metabolism[4,6–8]. Iron must be absorbed from the diet and vertebrate enterocytes absorb iron via Divalent Metal Transporter 1 (DMT1), followed by transportation across the basolateral membrane via ferroportin[9]. This allows iron to be loaded onto transferrin, a systemic iron transporter that accepts one or two ferric iron ions. Vertebrate transferrins reach target cells via the bloodstream, where they are endocytosed upon binding to the transferrin receptor (TFR1).

Although ferritin, a protein that forms iron-storing nanocages, is primarily cytosolic in humans, it is also found in the serum[10]. The presence of serum ferritin suggests a role in systemic iron trafficking, iron sequestration to protect against iron overload, or pathogen response. While it remains unclear whether ferritin directly participates in systemic iron trafficking, Scara5, TIM-2, and TFR1 receptors have been implicated in its endocytosis in human and mouse cells[11–13]. Cells may also retrieve iron from ferritin under iron-deficient conditions through ferritinophagy[14], a specific form of autophagy that targets ferritin to lysosomes for degradation[14,15].

In *Drosophila*, the prothoracic gland (PG) requires substantial amounts of iron to produce the steroid hormone ecdysone[16]. Most mammalian and insect steroidogenic enzymes require heme or Fe-S clusters as prosthetic groups[17,18], such as the cytochrome P450s (P450s). The PG, together with two other endocrine glands, forms the ring gland (RG). In the PG, a series of at least seven enzymes – referred to as the "Halloween" enzymes – synthesize α-ecdysone (a prohormone) from a suitable sterol precursor[19,20]. Once α-ecdysone is delivered to target tissues, a final conversion step generates 20-Hydroxyecdysone (20E),

[1]University of Alberta, Faculty of Science, Edmonton, Alberta T6G 2E9, Canada. [2]Stanford Synchrotron Radiation Lightsource SLAC National Accelerator Laboratory, 2575 Sand Hill Road, Menlo Park, CA 94025, USA. ✉e-mail: kingjone@ualberta.ca

one of the main biologically active forms of ecdysone[19–21]. The Halloween genes encode six P450 enzymes (which bind heme), a single short-chain dehydrogenase/reductase (does not require iron), and a Rieske electron oxygenase that harbors an Fe-S cluster[19,22]. At least three abundant PG-specific Halloween enzymes require high levels of heme and iron, contributing to the PG's iron-rich composition[20]. These mass-produced enzymes produce ecdysone pulses that govern developmental transitions such as molts and puparium formation. Heme is cytotoxic and hydrophobic and cannot be stored as such. Therefore, iron import and heme production must be tightly synchronized with the synthesis of Halloween enzymes, raising the question as to how iron demand is coordinated with steroid hormone production.

In this study we show that the Ecotropic viral integration site 5 (Evi5) is a hitherto unidentified regulator of vesicular iron transport in *Drosophila*, and critical for maintaining iron uptake from the hemolymph, the equivalent of insect blood. In insects, two hemolymph proteins, transferrin and ferritin, have been proposed to be the basis of systemic iron transport in insects, but direct evidence is lacking[23]. Flies mutant for ferritin are embryonic or early larval lethal[24]. Midgut-specific disruption of Fer1HCH function resulted systemic iron deficiency and in moderate or severe lethality, depending on whether iron was added to the diet or not, respectively[25]. As such, current data is compatible with the idea that ferritin delivers iron to target tissues, but an alternative view is that ferritin is critical for iron absorption from the diet and for the detoxification of peripheral tissues to avoid iron overload. Given that iron is an essential trace metal, it seems implausible that ferritin cannot be retrieved from the hemolymph by peripheral tissues, however, no studies have shown i) direct uptake of ferritin into insect cells or ii) that ferritin supplementation or hemolymph injections can rescue mutants with defects in iron metabolism.

*Drosophila* has three transferrin proteins, with only Tsf1 linked to iron homeostasis[26,27]. In contrast, humans have three transferrins and two transferrin receptor genes (*Tfr1* and *Tfr2*), which are part of the Transferrin Receptor/Glutamate Carboxypeptidase II (Tfr/GCP2) family. This family comprises seven genes encoding transmembrane proteins, but Tfr orthologs are exclusive to vertebrates[28], suggesting invertebrates use a different ancestral transferrin uptake mechanism. *Drosophila* Tsf1 null mutations are viable, suggesting alternative roles or multiple iron delivery systems. Tsf1 co-localizes with early endosome marker Rab5[26] (with the caveat that endocytosis is not needed for early endosome formation[29]), and is present on the gut surface[27], hinting at a role in iron transport. However, direct evidence of Tsf1 endocytosis is lacking, similar to the case with ferritin.

*Evi5* was initially identified as an integration site for Ecotropic viruses on the 5th mouse chromosome[30]. There are two human orthologs (hEvi5 and hEvi5-like) and one in *Drosophila* (Evi5). Evi5 has two conserved membrane-related domains, a TBC domain (Tre-2, Bub2p and Cdc16 proteins) and a coiled-coil domain. hEvi5 acts in cell cycle regulation, cytokinesis, and vesicle trafficking[30–34], and is considered an oncogene[32,34]. A series of genome-wide association studies (GWAS) implicate *Evi5* as a high-risk locus for MS[35–39]. This is potentially intriguing since MS patients display iron accumulation in the grey matter, which has led to the idea of a causal link between MS and iron[40,41].

*Drosophila* Evi5 regulates border cell migration, where it - similar to vertebrate Evi5 - acts as an effector protein of Rab11 via its TBC domain[30,31,33]. Rab proteins alternate between an active GTP-bound and an inactive GDP-bound form to regulate vesicle trafficking. The exchange of GDP with GTP is catalyzed by guanine nucleotide exchange factors (GEFs)[42], whereas GTPase-activating proteins (GAPs) inactivate Rab proteins by accelerating their inherently low intrinsic GTP to GDP hydrolyzation activity[42]. Remarkably, ~15% of human cancers are associated with mutations in genes encoding GEF and GAP proteins[43]. 173 human and 64 *Drosophila* genes encode GAP proteins acting on members of the Ras superfamily, of which 51 human and 27 *Drosophila* GAPs are Rab-specific due to their TBC domains[33,43].

In this study, we show that PG-specific loss of *Evi5* function substantially affects the number, size and morphology of cellular vesicles and blocks cellular iron trafficking of iron-loaded Tsf1. Furthermore, we found that Evi5 likely plays a role in ferritin trafficking, as Evi5 physically interacts with the ferritin heavy chain (Fer1HCH) both in vivo and ex vivo. Taken together, this study identified novel roles for Evi5 in directing iron-containing vesicles to target organelles, demonstrated unequivocally that Tsf1 is utilized in *Drosophila* for systemic (tissue-to-tissue) iron transport, and provided compelling, albeit indirect, evidence suggesting a contributory role for ferritin in this process.

## Results

### PG-specific loss of Evi5 function causes porphyria and developmental delays

To identify new components of iron metabolism and iron transport, we conducted an RNAi screen in the *Drosophila* PG, a tissue with unusually high iron demands. Specifically, we selected 803 RNAi lines that showed larval lethality in a prior PG-specific screen[44], and searched for the presence of red autofluorescing PGs, which is indicative of heme precursor build-up due to incomplete heme synthesis[16]. This phenotype is equivalent to the human porphyrias, which are characterized by mutations affecting heme biosynthesis, and the accumulation of toxic heme precursors known as protoporphyrins. Porphyrins fluoresce in a bright red when exposed to oxygen and UV light[45–47], but are usually short-lived and quickly converted to the next intermediate. However, mutations impairing heme biosynthesis can lead to insufficient clearance of protoporphyrins, resulting in their accumulation and causing red autofluorescence. The accumulation of these protoporphyrins is exacerbated due to the strong induction of the *Aminolevulinate synthase* (*ALAS*) gene, which encodes the rate-limiting enzyme of the heme biosynthetic pathway in mammals[48] and in *Drosophila*[16]. Our autofluorescence-based PG-specific RNAi screen yielded 21 genes, encompassing 16 genes with no known links to iron, including Evi5 and AGBE[16].

When *Evi5* function was targeted via PG-specific RNAi (using the PG-specific "phm22-Gal4" driver, hereafter referred to as "PG >", and "Inverted Repeat" = superscript "IR" for RNAi lines), the resulting line (*PG>Evi5[IR1]*) displayed enlarged, red autofluorescing PGs (Fig. 1A). This phenotype was comparable to depleting the penultimate heme biosynthetic enzyme, *PPOX*, in the PG (Fig. 1A), as well as targeting AGBE, IRP1A and mitoNEET, all three of which act together to control cellular iron homeostasis[16]. This suggested a role for Evi5 in iron or heme biology, but no reports have linked Evi5 to either.

We confirmed the *Evi5*-RNAi phenotype with a second, non-overlapping RNAi line (*PG>Evi5-RNAi[TRIP]*, hereafter named *PG>Evi5[IR2]*) (Fig. 1A–C). Overall, *PG>Evi5[IR2]* showed comparable but weaker results, with less tissue enlargement and less autofluorescence (Fig. 1A). To validate the *Evi5*-RNAi phenotypes with a different methodology, we used CRISPR/Cas9 to generate a conditional mutant line, *Evi5[FRT]*, where we replaced the endogenous transcription unit with a version flanked with Flippase Recombination Target (FRT) recognition sites (Fig. S1A). This *Evi5[FRT]* knock-in allele encoded C-terminal Flag epitopes (3x Flag, Fig. S1). When we excised the *Evi5[FRT]* allele via PG-specific Flippase (*PG > FLP;Evi5[FRT]*), we observed red autofluorescening PGs and enlarged ring glands similar to the *Evi5*-RNAi phenotypes (Fig. 1A). PG-specific excision of *Evi5* resulted in a significant reduction in Evi5 protein levels in ring gland (Fig. S1B) samples, as evidenced by Western blot analysis (Fig. S1C).

A hallmark of acute porphyria attacks (e.g., in acute intermittent porphyria) is the strong induction of the *Alas* gene. We, therefore, tested *Alas* expression levels in ring gland samples from the three *Evi5*-loss-of-function lines, for which we observed 28-, 22- and 8-fold *Alas* upregulation in *PG>Evi5[IR1]*, *PG >FLP;Evi5[FRT]*, and *PG>Evi5[IR2]* animals, respectively (Fig. 1B). This confirmed that heme biosynthesis was impaired in Evi5-depleted PGs. In addition to the red autofluorescence,

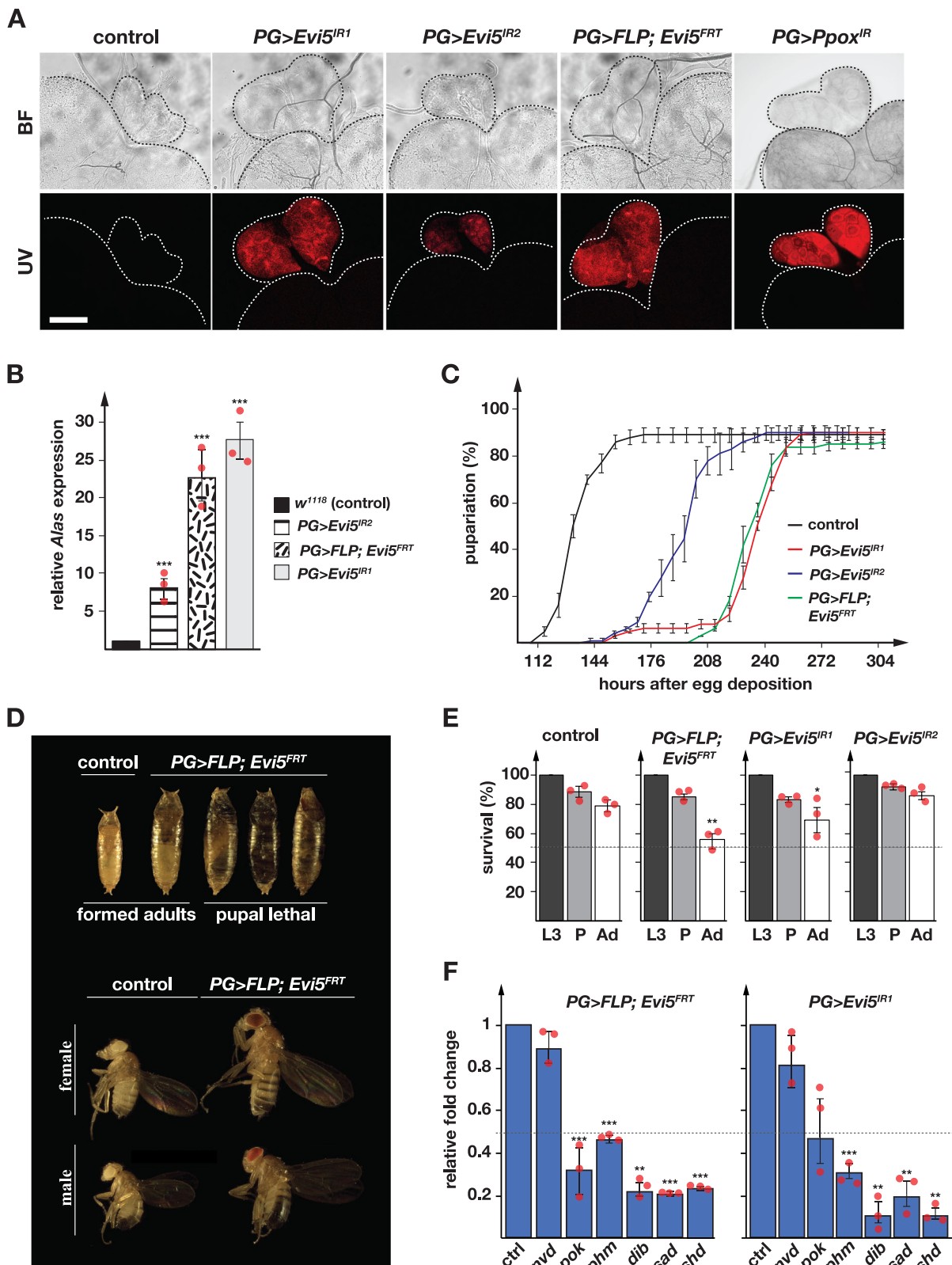

PG-specific depletion of Evi5 led to substantial developmental delays, resulting in a prolonged third instar larval (L3) stage with animals reaching pupariation 3-5 days later than normal (Fig. 1C). Delays prior to L3 were minimal, or absent. These developmental delays resulted in longer feeding times and correspondingly larger larvae, pupae and adults compared to controls (Fig. 1D). Loss of Evi5 function also resulted in moderate pupal lethality (20–45%, Fig. 1E), suggesting that

Evi5 function is critical (in the PG) during L3 and pupal stages. Since Fe-S and heme are co-factors of ecdysteroidogenic enzymes, we tested the expression levels of six Halloween genes in the PG. For this, we compared *PG>Evi5^IR1*, *PG >FLP;Evi5^FRT* and controls via qPCR, which showed a twofold and up to tenfold reduction of Halloween transcripts in Evi5-depleted samples, with the exception of *neverland* (*nvd*), which appeared unaffected (Fig. 1F).

**Fig. 1 | PG-specific disruption of *Evi5* impairs heme and steroid hormone biosynthesis. A** Presence of red autofluorescence in larval ring glands. Brain-ring gland complexes were dissected from *PG >w[1118]* (control), *PG>Evi5[IR1]*, *PG>Evi5[IR2]*, *PG >FLP;Evi5[FRT]* and *PG>Ppox[IR]* larvae and exposed to brightfield light (BF) or ultraviolet light (UV). "IR" stands for "inverted repeat" and refers to the RNAi lines, whereas "FRT" indicate the presence of Flippase (FLP)-recombinase targets, which are here used to conditionally excise the *Evi5* transcription unit via FLP. Dotted lines highlight the ring gland and the adjacent section of the CNS. Scale bar = 250 μm. **B** qPCR analysis of *ALAS* transcripts in ring gland samples. Fold changes calculated with ddCT method and error bars represent 95% confidence intervals, both are based on three biological replicates (each tested in triplicate). **C** Developmental timing analysis. Error bars represent standard error (SE) based on three replicates. Controls are *PG > w[1118]*. **D** Pupal and adult sizes of *PG >FLP;Evi5[FRT]* animals. Longer

feeding times due to developmental delays (Fig. 1C) resulted in larger final body sizes. A significant proportion of pupae did not progress to the adult stage (Fig. 1E) and displayed aberrant pupal development. **E** Survival analysis of 3rd instar larval (L3), pupae (P) and adults (Ad) upon PG-specific disruption of *Evi5* function compared to controls. The dotted line indicates 50% survival. Error bars show standard error. Three biological replicates tested (each = 50 individuals). **F** qPCR analysis testing the expression of ecdysone biosynthetic genes in ring gland samples. ctrl control, nvd neverland; spok spookier; phm phantom; dib disembodied, sad shadow, shd shade. Error bars are 95% confidence intervals. Fold changes and error bars as described in (**B**). Asterisks in (**B**, **E** and **F**) denote significance level of two-sided Student's *t*-test between controls and experimental line (***$p < 0.001$). In (**B**, **C**, **E**, **F**) center of error bars denote the average. Source data are provided in the source data file.

## Evi5 depletion reduced cellular iron levels

The expression profiles for *Alas* and the Halloween genes (Fig. 1B, F) suggested that Evi5 is required for heme and ecdysone biosynthesis. To further validate this, we reared *PG >FLP;Evi5[FRT]* animals on media supplemented with iron (in the form of ferric ammonium citrate = FAC), hemin (a heme analog = $Fe^{3+}$-protoporphyrin IX), a porphyrin containing zinc rather than iron (Zn-protoporphyrin IX = ZnPP), and 20E. We hypothesized that the phenotypes (autofluorescence, developmental delays and survival) associated with PG-specific *Evi5*-disruption may be rescued by dietary compounds that compensate for defects in these animals. We first tested whether these compounds affected the red autofluorescence (Fig. 2A) and *Alas* expression in *PG >FLP;Evi5[FRT]* (Fig. 2B). Interestingly, iron in the form of FAC supplementation had only minor effects. In contrast, hemin as an iron source resulted in a substantial rescue of both the autofluorescence and *Alas* expression. Our earlier work established that feeding FAC was effective in rescuing the autofluorescence, *Alas* expression and survival of *IRP1A*, *AGBE*, and *mitoNEET* loss-of-function lines[16]. Since FAC-feeding failed in this case, we surmised that Evi5 is required for the uptake or distribution of diet-derived iron. To rule out that the hemin rescue was caused by the protoporphyrin ring rather than iron, we also tested ZnPP, which is identical to hemin except for the bound metal. Dietary supplementation with ZnPP could not rescue Evi5 depletion. To ensure that ZnPP was intact, we fed hemin and ZnPP to *PG>Ppox[IR]* larvae, where heme production is disrupted at the penultimate step, causing larval and pupal lethality. Both hemin and ZnPP rescued *PG>Ppox[IR]* animals to adulthood (Fig. 2C), demonstrating that i) ZnPP is functional, ii) ZnPP is not broken down in the gut and instead transported intact to the prothoracic gland, and iii) providing protoporphyrins lacking iron was not sufficient to rescue Evi5-depleted animals. Taken together, this suggested that hemin passes through the gut intact, which is consistent with our finding that Evi5-impaired animals are rescued by dietary hemin but not FAC. 20E feeding also rescued both the autofluorescence and *Alas* expression, but not as effectively as hemin. This suggested that Evi5 depletion affected ecdysone production by disrupting the heme pathway.

The above-mentioned compounds had corresponding effects on the developmental delay of *PG >FLP;Evi5[FRT]* animals. ZnPP was the least effective in reducing the development delay (~4 h), followed by FAC (~24 h). In contrast, hemin and 20E supplementation were effective, reducing delays by 88 and 96 h, respectively (Fig. 2D). Finally, we tested whether these compounds could rescue the pupal lethality of *PG >FLP;Evi5[FRT]* animals (Fig. 2E). As expected, ZnPP-supplementation did not improve survival rates. Somewhat surprisingly, FAC improved adult survival, possibly because dietary iron was beneficial for pupal development but less effective for shortening larval development. Hemin supplementation worked best, which showed a nearly complete rescue, followed by ecdysone administration.

To directly assess the presence of iron in PGs isolated from controls and *PG >FLP;Evi5[FRT]* mutants, we performed X-ray fluorescence (XRF) at the Stanford Synchrotron Radiation Light Source, which

allowed us to create an iron distribution map of brain-ring gland complexes (BRGCs) (Fig. 2F). Since the developmental delays and the autofluorescence were L3-specific, we dissected BRGCs at different time points from wandering L3. The mean iron level was calculated for each sample, and the RGs and the CNS were assessed separately. This approach showed statistically significant iron depletion in 60 and 74 h old RGs, but not the CNS, from *PG >FLP;Evi5[FRT]* larvae, when compared to earlier time points (Fig. 2G). Importantly, there were no substantial iron concentration changes in the CNS of these animals. In control animals, we observed no significant iron concentration changes in the RGs or in the CNS. These developmentally late iron depletion phenotypes were consistent with our finding that *PG >FLP;Evi5[FRT]* animals displayed significant pupal lethality and demonstrated that Evi5 was critical for maintaining proper iron levels in the PG.

## Loss of Rab5 and Rab11 function causes porphyria-like phenotypes

Rab proteins are small GTPases that are critical for regulating membrane traffic, including endosome recycling, exocytosis, and Golgi/endosome transport processes. In vertebrates, Rab5 and Rab11 are critical for cellular cargo delivery via endocytosis and exocytosis, including iron[49,50]. Both Rab5 and Rabb11 have been linked to the endocytosis of transferrin, since Rab5 co-localized with the transferrin receptor[51], whereas Rab11 was required for transporting transferrin back to the cell surface[52]. Evi5 functions as a GTPase-activating protein GAP for Rab11 and binds directly to the GTP-bound form of Rab11 to coordinate vesicular trafficking[31]. Iron enters cells via endocytosed clathrin-coated vesicles harboring transferrin-bound TfR1 complexes. Rab5-positive early endosomes (EE) are then acidified, which facilitates the removal of $Fe^{3+}$ from transferrin, and subsequently sorted into Rab11-containing recycling endosomes (RE) that return TfR1 to the plasma membrane. However, Rab11 and TfR1 co-localize only to some degree[53], suggesting that Rab11 has additional roles or that other Rab proteins can import TfR1-containing vesicles.

Next, we examined whether *Drosophila* Rab5 and Rab11 played a role in the vesicular distribution of iron and whether Evi5 was necessary for proper Rab5/Rab11 function, and conducted PG-specific depletion of *Rab5* and *Rab11*. *PG>Rab5[IR]* animals displayed a ~30 h delay (Fig. 3A, G), but survived to adulthood. In contrast, only 5% of the *PG>Rab11[IR]* population formed pupae, with no animals forming adults (Fig. 3B, G). Both *PG>Rab5[IR]* and *PG>Rab11[IR]* animals had slightly enlarged RGs with weak red autofluorescence, but it took longer for these phenotypes to develop as these were seen in ~65–70 h old L3 for *PG>Rab5[IR]* larvae and in 4-day-old *PG>Rab11[IR]* L3 (compared to Evi5-loss-of-function L3 at 44 h) (Fig. 3C). Consistent with this, *Alas* was upregulated 2- and 4-fold in *PG>Rab5[IR]* and *PG>Rab11[IR]* animals, respectively (Fig. 3D), which is 5-10 times lower than in Evi5-loss-of-function animals. When we examined the expression of *Rab5* and *Rab11* in the RG of *PG >FLP;Evi5[FRT]* (Fig. 3E), we found that *Rab11* expression was 5-fold reduced, whereas *Rab5* expression was unaffected. This suggested that *Rab11* expression is linked to Evi5 activity,

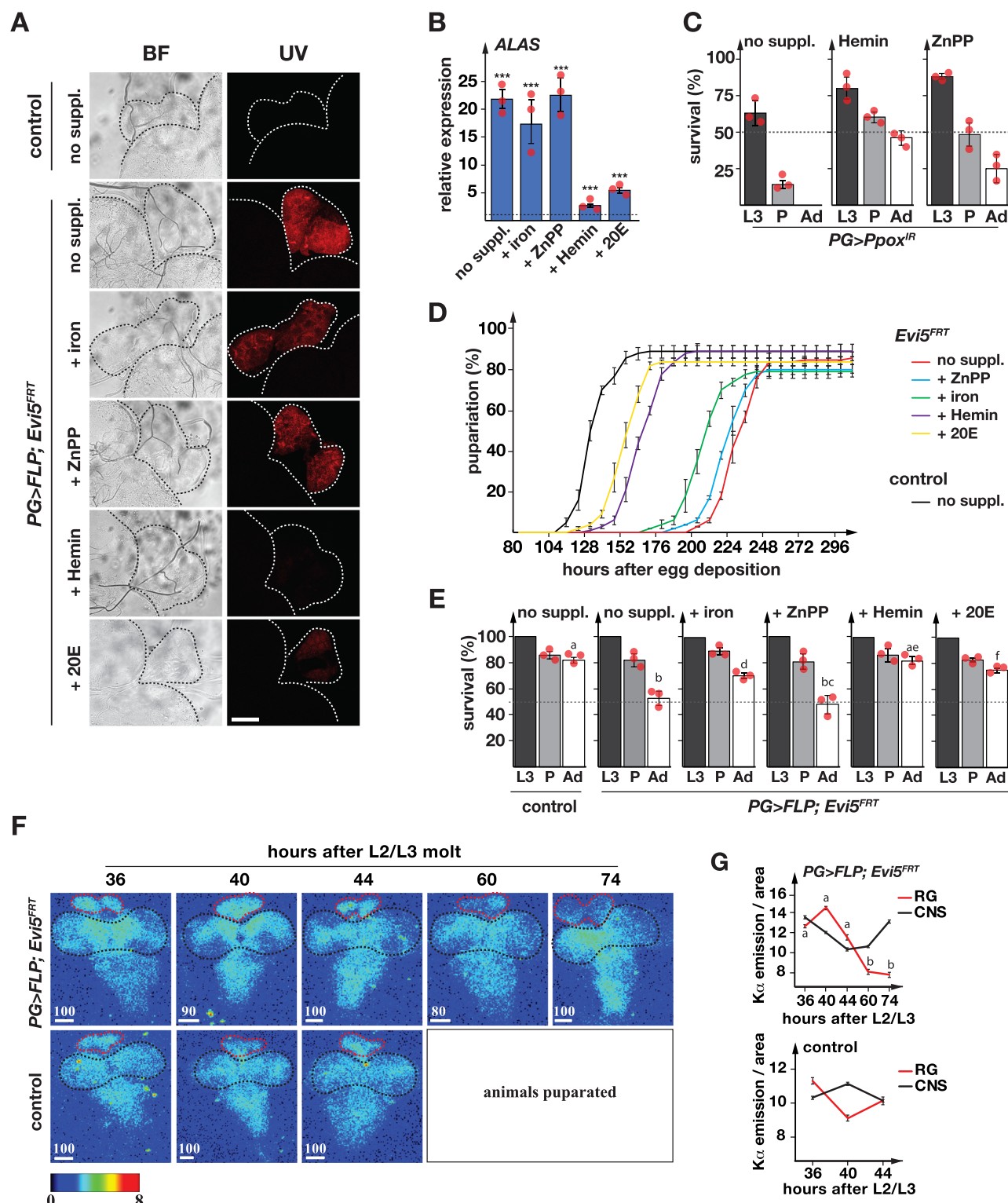

whereas *Rab5* is not transcriptionally regulated in this fashion. To test whether Evi5 function was indeed dependent on Rab11, we attempted to rescue *Evi5* loss-of-function phenotypes by generating two transgenes that we subsequently crossed into *PG>FLP;Evi5^FRT* animals (Fig. 3F). One transgene (*UAS-Evi5^WT-mVenus*) encoded wild-type Evi5 tagged with mVenus, whereas the other variant (*UAS-Evi5^R160A-mVenus*) was identical except for a point mutation in the GAP domain at position #160 that caused Evi5 to be catalytically inactive, as previously demonstrated[33]. The wild-type Evi5 transgene rescued the autofluorescence, whereas catalytically inactive Evi5 failed to do so

(Fig. 3F), demonstrating that Evi5's GAP domain was required for this process.

Next, we tested the uniqueness of the phenotypes associated with loss of Evi5, Rab5 and Rab11 function, since it was possible that these phenotypes were a common feature of PG-specific depletion of Rab or GAP proteins. To this end, we carried out two small RNAi screens that targeted these gene families (Fig. 3G, H). The first screen targeted the 33 known *Drosophila Rab* genes in the PG. In *Drosophila*, there are 27 Rab proteins that have unequivocal vertebrate orthologs and six RabX proteins with lower homology[54,55]. PG-specific depletion of the 33 *Rab*

**Fig. 2 | PG-specific depletion of Evi5 causes iron-deficiency. A** Red auto-fluorescence in ring glands from $PG > w^{1118}$ (control) and $PG > FLP;Evi5^{FRT}$ larvae reared on a normal diet (no suppl. = no supplement), or on media supplemented with FAC (iron), Zinc-Protoporphyrin IX (ZnPP), Hemin and 20-Hydroxyecdysone (20E). The dotted lines show boundaries of the brain and the RG under UV and brightfield (BF) light. Scale bar = 250 μm. **B** Expression of *ALAS* in RGs isolated from $PG > FLP;Evi5^{FRT}$ larvae relative to the control (shown as dotted line = 1) after being raised on different foods. Fold changes calculated with ddCT method and error bars denote 95% confidence intervals, both are based on three biological replicates (each tested in triplicate). The center of the error bars denotes the average. Asterisks are P-values obtained via a two-sided, paired Student's t-test (***$p < 0.001$). **C** Survival rates of $PG>Ppox^{IR}$ on normal diet (no suppl.), Hemin- and ZnPP-supplemented media. **D** Pupariation timing of $PG > FLP;Evi5^{FRT}$ animals when reared on different media. **E** Pupal lethality of $PG > FLP;Evi5^{FRT}$ animals when reared on different media. L3: 3rd instar larvae, P: pupae, Ad: adults. Significance letters are based on P-values obtained via a two-sided, paired Student's t-test (***$p < 0.001$). In C-E, error bars represent standard error, with the center denoting the average. In (**C–E**), three biological replicates were used, with each sample consisting of 50 individuals. **F** X-Ray Fluorescence Microscopy (XRF) images of BRGCs from control and $PG > FLP;Evi5^{FRT}$ animals. At the 60 and 74 h, the control animals underwent puparium formation and are no longer comparable to the $PG > FLP;Evi5^{FRT}$ larvae. **G** Quantification of iron distribution in ring glands (red dotted line) vs. CNS (black dotted line) from control and $PG > FLP;Evi5^{FRT}$ animals. Image scale bars represent 80–100 μm. The color scale bar is log2-based and represents tissue iron levels. Significance in E and G is based on ANOVA and letter differences indicate a two-sided *P*-value of $p < 0.05$. Error bars denote standard deviation of number pixels in selected areas, with the center denoting the average. Source data are provided in the source data file.

genes yielded only two lines with red autofluorescence in the PG, namely the aforementioned Rab5- and Rab11-RNAi lines. Three of the lines caused L3 lethality, namely $PG>Rab11^{IR}$, $PG>Rab4^{IR}$ and $PG>RabX6^{IR}$. 20 lines exhibited developmental delays, of which Rab5-RNAi had the strongest delay (32 h), with the remaining lines exhibiting delays between 4 and 16 h. Nine lines displayed no apparent pheno-types. In conclusion, the porphyria-like phenotypes were specific to Rab5 and Rab11 and not a common feature of impairing Rab proteins.

Since Evi5 acts as a Rab-GAP by stimulating the GTPase activity of Rab11, PG-specific Evi5-depletion should increase Rab11-GTP levels. Thus, a constitutively active form of Rab11 (Rab11$^{CA}$) should phenocopy – at least in part – the loss of Evi5 function. When we monitored the developmental progression of $PG>Rab11^{CA}$ animals, we saw a devel-opmental delay of ~32 h (Fig. 3F), but we did not observe any red autofluorescence.

In the second RNAi screen, we used the same strategy to knock down all known GAPs, and we also included known GEF proteins for Rab5 and Rab11 (Fig. 3H). The second RNAi screen allowed us to test whether i) Evi5 was the only GAP protein associated with heme defi-ciency phenotypes and ii), the PG-specific depletion of other Rab5 and Rab11 effectors (i.e., GEF proteins) would cause autofluorescence. In *Drosophila*, there are 27 known and presumed Rab-GAPs[33], as well as five Rab5-GEFs and three Rab11-GEFs. This strategy showed that except for Evi5, none of the remaining 25 RNAi lines produced red auto-fluorescence. With respect to developmental delays, Muscle-specific protein 300 kDa (Msp300) was the only Rab-GAP RNAi line alongside Evi5-RNAi that exhibited a significant effect (Msp300: ~50 h vs. Evi5: 70 and 120 h) (Fig. 3H).

Interfering with the function of all Rab5 GEFs caused no significant phenotypes. Only the PG-specific loss of *Crag* function showed a ~60 h delay at the L3 stage and ~20% lethality at the pupal stage. However, Crag-RNAi did not result in autofluorescing RGs (Fig. 3H). Crag func-tion differs from Evi5, as the former physically interacts with GDP-Rab11 to exchange GDP to GTP, resulting in the activation of Rab11. Evi5, on the other hand, physically interacts with GTP-Rab11 via its TBC domain to induce GTP to GDP conversion, which inactivates Rab11[30,31,33].

Taken together, these results suggest that Rab5, Rab11 and Evi5 function in different aspects of the same pathway and impinge on the heme biosynthetic pathway, consistent with the idea that these pro-teins are critical for the vesicular transport of iron.

### Loss of Evi5 function disrupts vesicle transport

Vertebrate Rab5 is involved in the internalization of holo-Tsf (Tsf-Fe$^{3+}$) into early endosomes (EE), and Rab11 governs the return of iron-free apo-Tsf to the plasma membrane as recycling endosomes (RE)[49]. We hypothesized that vesicle recycling would be blocked in the absence of functional Evi5, thus entrapping vesicles in the PG and disrupting iron trafficking to an extent that would impair heme, iron-sulfur cluster and ecdysone synthesis. To test this, we immunolabelled hepatocyte-responsive serum phosphoprotein (Hrs), a marker for early endo-somes (EE), as it co-localizes with Rab5 on EE[56]. BRGCs from $PG > FLP;Evi5^{FRT}$ and $PG>Evi5^{IRI}$ larvae showed 3-4 times higher Hrs levels than controls (79.4 and 63.5 *vs.* 22.5 signal density), indicating EE accumulation (Fig. 4A–C). This increase of Hrs signal density was not due to a higher number in PG cells, which were comparable between controls and Evi5-loss-of-function PGs (Fig. 4C).

Next, we used Transmission Electron Microscopy (TEM) to analyze vesicle density (Fig. 4D, E). We observed a ~ 2-fold increase in vesicle abundance in PGs isolated from $PG > FLP;Evi5^{FRT}$ larvae compared to controls (Fig. 4F). Almost all vesicles were larger compared to controls, with an average ~6-fold increase in diameter. (Fig. 4F). Furthermore, we observed abnormal vesicles with low-density lumen and vesicles with dark halos around or inside the vesicle lumen that were absent in controls (Fig. 4D, yellow arrows). The abnormal dark vesicles appeared to be defect lysosomes (Fig. 4E), and they resembled electron-dense iron-containing lyso-somes described in mice and humans[57,58]. TEM also confirmed the increased size of Evi5-depleted PG cells. Lastly, as a second control, we performed TEM of brain cells isolated from controls and $PG > FLP;Evi5^{FRT}$ animals to ensure that abnormal vesicles were spe-cific to Evi5-depleted cells. As expected, we found no significant differences between control and mutant $Evi5^{FRT}$ brain cells (Fig. 4D). To complement the observed lysosomal phenotypes by a different assay, we stained PGs isolated from in $PG > FLP;Evi5^{FRT}$ larvae with Lysotracker blue, which revealed a 2.5-fold drop in lysosomal acidity (Fig. 4G, H). This suggested that Evi5 is required for lysosome for-mation, and that vesicles in an Evi5-depleted background do not properly progress to the lysosomal stage, consistent with the idea that this results in reduced iron release, since acidification is thought to release iron from transferrin.

In summary, we employed various strategies (Hrs immunostain-ing, TEM analysis and Lysotracker staining), which showed that dis-ruption of *Evi5* caused drastic changes in vesicle morphology and abundance, as well as lysosome formation.

### Evi5 is recruited to vesicles that contain transferrin proteins

Given that vesicular trafficking was impaired in $PG > FLP;Evi5^{FRT}$ animals, we used transferrin as a tool to examine intracellular transport of iron in PG cells. Despite the absence of a human TfR1 ortholog in *Droso-phila*, flies encode three transferrin proteins (Tsf1, Tsf2, and Tsf3). It was thus unclear whether fly transferrins are involved in iron delivery and whether they are endo- and exocytosed by cells. To examine this, we performed organ co-culture experiments with BRGCs and fat bodies, which allowed us to interrogate genotype combinations that cannot be obtained in vivo. Since Tsf2 and Tsf3 do not appear to have a major role in cellular iron uptake and homeostasis[27,59], we focused on Tsf1, for which some data suggested a similar role to mammalian transferrins[27,60]. Specifically, we tested whether the PG endocytosed fat body-derived Tsf1 and whether this depended on Evi5.

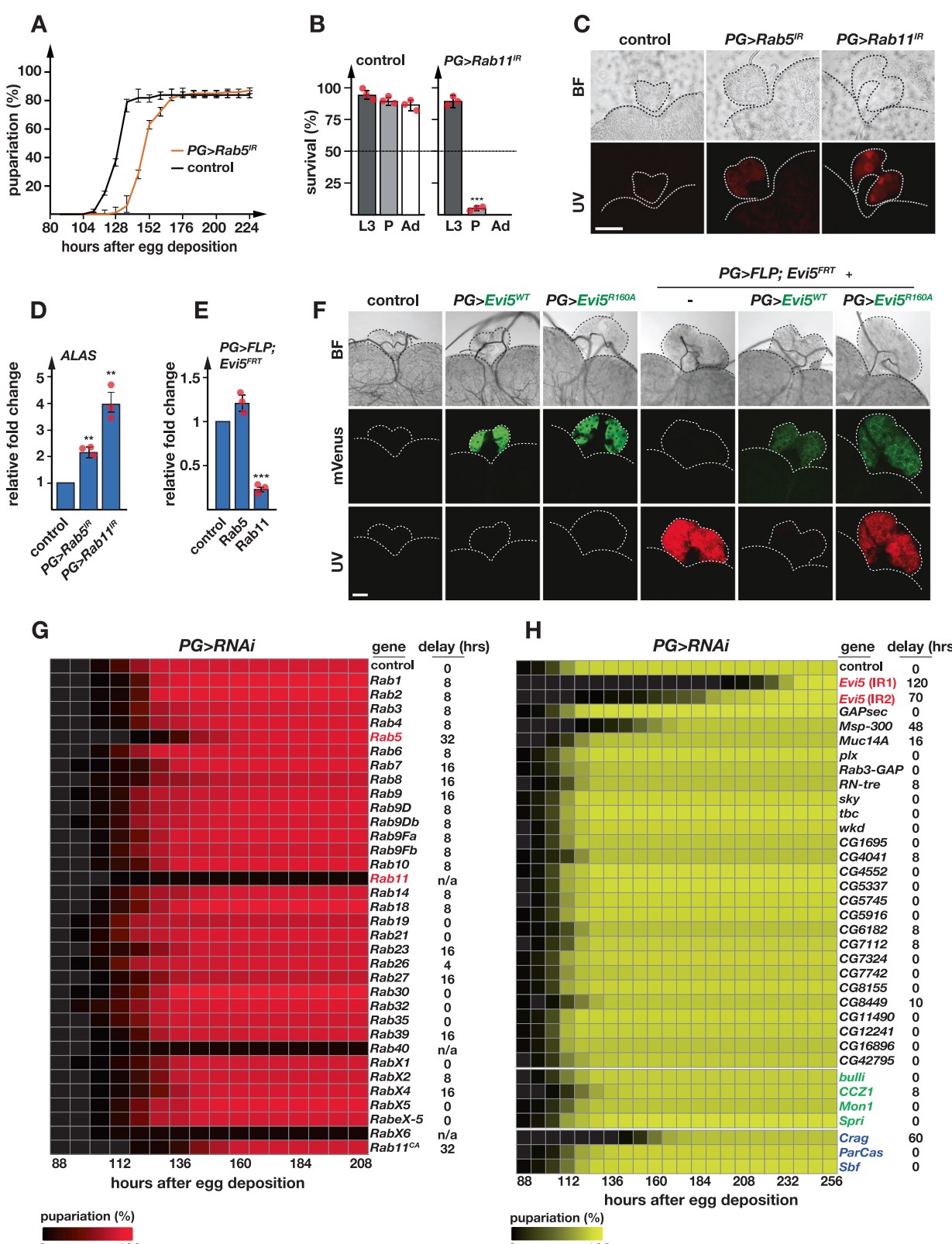

We first carried out an ex vivo organ co-culture experiment using BRGCs isolated from *PG>FLP;Evi5^FRT* larvae and fat bodies from a transgenic line expressing a GFP-tagged *Tsf1* gene, which mimics endogenous Tsf1 expression[61]. Studies in *Drosophila* and other insects demonstrated that Tsf1 is primarily expressed and released from the fat body[27]. Therefore, we used fat bodies as a source of Tsf1-GFP to test whether Tsf1 is secreted from the fat body and incorporated by co-

cultured BRGCs (Fig. 5A). Cultured *PG>FLP;Evi5^FRT* BRGCs exhibit stable red autofluorescence for up to four days; however, RGs displayed some morphological changes on the 4^{th} day (Fig. S1D). Data shown here are based on 24–48 h co-cultures. Expression of *Tsf1-GFP* was strong in cultured fat bodies and absent in controls that lacked the transgene (Fig. 5B). Tsf1-GFP levels increased when the culture medium was iron-deprived and decreased under iron-loaded conditions when

**Fig. 3 | PG-specific RNAi screen of Rab, Rab-GAP, Rab5-GEF and Rab11-GEF genes. A** Time required for control (*PG >w^{1118}*) and *PG>Rab5^{IR}* animals to reach pupariation. Error bars represent standard error of three biological replicates (each = 50), with the center indicating the average. **B** Survival analysis of *PG>Rab11^{IR}* animals. Error bars are standard error of the mean for three biological replicates (each = 50) and asterisks represent the P-value obtained by the two-sided Student's t-test (\*\*\**p* < 0.001). L3: 3^{rd} instar larvae, P: pupae, Ad: adults. IR = RNAi line. **C** Red autofluorescence of ring glands obtained from *PG>Rab5^{IR}* and *PG>Rab11^{IR}* animals. UV: ultraviolet light, BF: bright field. Scale bar = 250 μm. **D** qPCR analysis of the *ALAS* in ring glands of *PG>Rab5^{IR}* and *PG>Rab11^{IR}* animals. **E** Relative expression of *Rab5* and *Rab11* in ring glands isolated from *PG >FLP;Evi5^{FRT}* animals based on qPCR analysis. In (3**D**, 3**E**), fold changes were calculated with ddCT method and error bars denote 95% confidence intervals, both are based on three biological replicates

(each tested in triplicate). The center of error bars denotes the average. Asterisks are *P*-values obtained via a two-sided, paired Student's t-test (\*\*\**p* < 0.001). **F** PG-specific expression of *Evi5^{WT}-mVenus* and *Evi5^{R160A}-mVenus* cDNAs (green label) in either wild type or *Evi5* mutant (*PG > FLP;Evi5^{FRT}*) backgrounds. The dotted lines outline the ring gland and attached brain section under brightfield (BF), green (mVenus) and Red (UV) channels. Scale bar = 150 μm. **G. H** Heatmaps showing pupariation profiles for lines expressing PG-specific RNAi targeting *Drosophila* Rab, Rab-GAP, Rab5-GEF and Rab11-GEF genes. Shown is the percentage of animals that have reached puparium formation at a given time point. Gene names in red (*Evi5, Rab5* and *Rab11*) indicate the presence of red autofluorescing ring glands in the corresponding RNAi line. Rab11^{CA} is a constitutively active form of Rab11. Green gene names represent Rab5-GEFs (nucleotide exchange factors), whereas blue names represent Rab11-GEF genes. Source data are provided in the source data file.

compared to standard S2M medium, consistent with earlier reports[27]. We measured a ~ 4-fold difference in Tsf1-GFP levels between iron-rich and iron-depleted conditions (Fig. 5C). The increase of *Tsf1* expression can be interpreted as a compensatory response to maximize iron-binding when the iron is scarce, but we favour the idea that Tsf1 secretion is reduced when iron levels are low, resulting in an increase of fat body Tsf1 protein levels.

Next, we tested whether fat body-derived Tsf1-GFP was taken up by control PGs in a co-culture experiment (Fig. 5D, top). Remarkably, we detected GFP in the RG and the CNS of control BRGCs that lacked a GFP transgene, demonstrating i) that Tsf1-GFP was secreted from fat bodies and ii) that Tsf1-GFP was endocytosed by the PG and by brain cells (Fig. 5D). Remarkably, when we repeated the experiment with BRGCs isolated from *PG >FLP;Evi5^{FRT}* larvae, the resulting Tsf1-GFP signal was ~7-fold higher in PG cells (but not the brain) (Fig. 5D), consistent with the earlier observation that vesicles accumulated in *PG >FLP;Evi5^{FRT}* larvae (Fig. 4). We then manipulated iron levels in the culture medium. When we depleted iron via BPS, we observed a ~ 3-fold drop in Tsf1-GFP that reached the PG of *PG >FLP;Evi5^{FRT}* larvae, suggesting that there is a strong reduction of iron loading onto Tsf1 in fat bodies, with a concomitant decrease of iron-bound Tsf1 release into the medium. Increasing iron levels resulted in a ~ 5.8-fold increase of GFP in *PG >FLP;Evi5^{FRT}* PGs, comparable to regular medium. Importantly, the red autofluorescence in *PG > FLP;Evi5^{FRT}* could not be rescued by Tsf1-GFP uptake. One explanation was that Tsf1-GFP was not bound to iron in the first place, but it was also possible that Tsf1-derived iron was not delivered to its intracellular destinations due to the collapse of Evi5-mediated vesicle transport. To distinguish the two scenarios, we repeated the experiment with *PG > AGBE^{IR}* larvae, which also exhibit red autofluorescing PGs, however, this is due to impaired iron homeostasis rather than a defect in vesicular transport[16]. Importantly, the lethality seen in loss-of-*AGBE*-function animals can be rescued by adding iron to the diet[16], raising the possibility that iron delivery via Tsf1-GFP could rescue the red autofluorescence phenotype of co-cultured *PG >AGBE^{IR}* RGs. Remarkably, providing Tsf1-GFP in the form of co-cultured fat bodies resulted in a dramatic rescue of the red autofluorescence displayed by *PG >AGBE^{IR}* RGs (Fig. 5D), with a concomitant ~9-fold reduction of *Alas* induction, compared to a 1.5-fold decrease in *PG >FLP;Evi5^{FRT}* ring glands (Fig. 5E). In addition, Tsf1-GFP levels were substantially lower than in Evi5-depleted larvae and comparable to controls, indicating normal vesicle traffic in AGBE-depleted PGs (Fig. 5D). Notably, the red autofluorescence in RGs from *PG > AGBE^{IR}* larvae remained high when the iron chelator BPS was added, with a corresponding reduction in Tsf1-GFP levels.

Next, we asked whether the rescue of red autofluorescence depended on Tsf1 function. To test this, we co-cultured fat bodies of control and *Tub-GAL4>Tsf1^{IR}* animals with *PG >AGBE^{IR}* and *PG >FLP;Evi5^{FRT}* BRGCs. In the presence of wild-type fat bodies, the autofluorescence in *PG >AGBE^{IR}* PGs was nearly gone. However, when

Tsf1-depleted fat bodies were used, the red autofluorescence persisted in *PG >AGBE^{IR}* ring glands (Fig. 5F). As expected, no rescue was seen in *PG >FLP;Evi5^{FRT}* RGs.

Next, we asked whether co-culturing Tsf1-GFP fat bodies increased iron levels in the RG from control and *PG >FLP;Evi5^{FRT}* backgrounds (Fig. 5G). This approach revealed that iron concentrations increased 65-100% when Tsf1-GFP fat bodies were added. However, no significant iron increase was observed in *PG >FLP;Evi5^{FRT}* ring glands. This suggested that Tsf1-GFP accumulated in Evi5-depleted cells due to impaired vesicle recycling, disrupting iron uptake over time and ultimately reducing iron concentrations in the cell.

We then tested whether it was possible to observe co-localization of Tsf1-GFP with lysosomes by staining RGs in co-cultured BRGCs with Lysotracker Blue (Fig. S1E). However, we did not observe co-localization in either control or *PG>FLP;Evi5^{FRT}* BRGCs. This is likely for three reasons. First, the pH in lysosomes ( ~ 4.5) is just outside the pH working range of most commonly used GFP variants (pH 4.8–8.0)[62]. Another issue exists in *PG >FLP;Evi5^{FRT}* samples, where lysosomes are either less abundant or acidified to a lesser degree, as mentioned earlier (Fig. S1E and Fig. 4G). Finally, previous work suggests that the amount of transferrin in the lysosomal compartment is small at any given time due to the continuous recycling of the vesicles[63].

As an alternative approach, we generated transgenic lines producing C-terminally Flag-tagged Tsf proteins (*UAS-Tsf1-3XFlag, UAS-Tsf2-3XFlag* and *UAS-Tsf3-3XFlag*). Tsf proteins harbor an N-terminal secretion signal peptide that will be cleaved upon protein secretion (Fig. S2A–C), necessitating a C-terminal tag. We next expressed each of the *Tsf* transgenes in an *Evi5*-depleted background (*PG >FLP;Evi5^{FRT};UAS-Tsf1,2&3-3XFlag*). To test whether Evi5-depletion would cause the accumulation of Tsf1-3, we immunostained BRGCs and conducted Western blots with hemolymph samples (Fig. S3). This strategy demonstrated that the loss of Evi5 function caused a strong accumulation of all three Tsf proteins (Fig. S3A). Since Tsf1-3 proteins were only weakly detectable in wild-type RGs, we concluded that all three proteins are normally secreted. Consistent with this, the Western blot analysis identified Tsf proteins in the hemolymph of control animals but not in those with PG-specific Evi5-depletion (Fig. S3B). Taken together, these data demonstrate that Evi5 is essential for vesicular trafficking of the transferrin proteins, both for the uptake of iron-loaded Tsf1 as well as the secretion of newly formed Tsf1 proteins (Fig. S3C).

We then addressed whether Evi5 co-localized with Tsf proteins in S2 cells. Specifically, we co-transfected cells with mCherry-tagged Evi5 (Evi5-mCherry) and eGFP-tagged Tsf1, Tsf2 and Tsf3. For Tsf1-3, we deleted DNA encoding the signal peptide and added a GFP-tag (Tsf1^{ΔSP}-eGFP, Tsf2^{ΔSP}-eGFP and Tsf3^{ΔSP}-eGFP). This approach showed that Evi5-mCherry was mainly recruited to Tsf-positive vesicles derived from the secretory pathway, which was the case for all three Tsf proteins (Fig. S4).

Taken together, these data demonstrate Evi5's involvement in vesicular iron transport. Tsf1 delivers iron to target tissues and is sufficient to rescue the porphyria phenotype in AGBE-depleted PGs, but

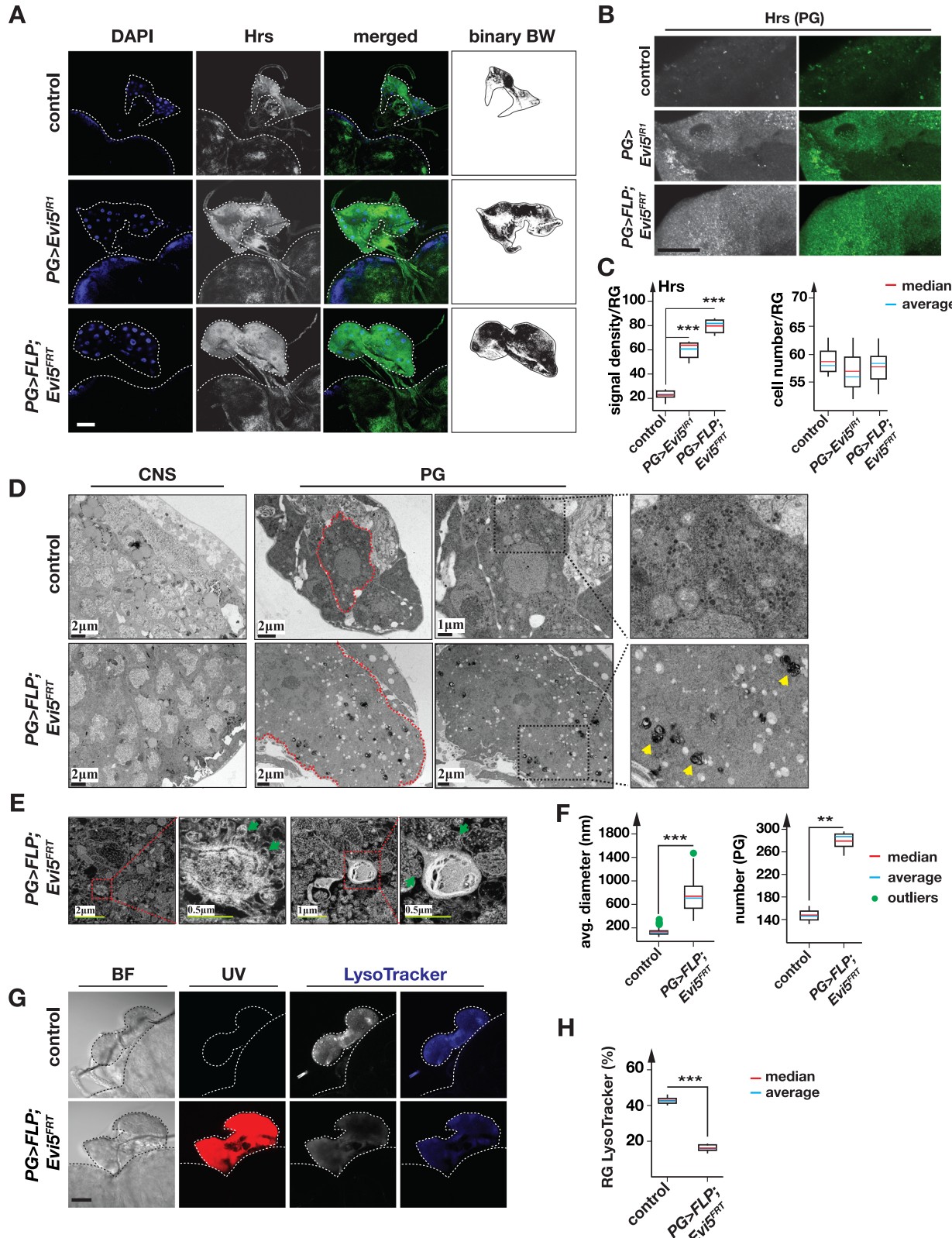

not when Evi5 is impaired. Loss of Evi5 function still allows for the uptake of Tsf1, but the subsequent vesicular transport is disrupted, causing a failure to deliver iron to target organelles.

## Evi5 interacts physically with ferritin

To further validate the role of Evi5 in iron transport, we carried out co-immunoprecipitation experiments to identify physical interaction

partners of Evi5. For this, we used the *Evi5<sup>FRT</sup>* line as well as S2 cells transfected with plasmids producing Flag-tagged Evi5. Both approaches were followed by MALDI-TOF mass spectrometry (MS) to identify co-immunoprecipitated proteins (IP-MS). As controls, we used wild-type flies that lacked Flag-tagged proteins for the *Evi5<sup>FRT</sup>* line and Flag-tagged eGFP as the control for the S2 cell experiment. For either approach, we removed all proteins from the final list if they appeared

**Fig. 4 | Loss of *Evi5*-function of causes accumulation of early endosomes.**
**A, B** Immunodetection of hepatocyte responsive serum phosphoprotein (Hrs) in ring glands of *PG >w^{1118}* (control), *PG >Evi5^{IR1}* and *PG >FLP;Evi5^{FRT}* animals, respectively. The blue channel shows DAPI to stain DNA, whereas the grey and green channels represent Hrs. Binary black and white (BW) images were used for Hrs quantification shown in (**C**). Scale bar = 250 μm. **C** Box plots showing Hrs signal density (six biological samples were tested and normalized to RG area, left), and ring gland cell numbers (10 biological samples were tested, right). Blue and red lines indicate average and median, respectively. Asterisks are P-values based on the two-sided Student's t-test (***$p < 0.001$). **D** Bright field TEM analysis of PG and brain cells in control and *PG >FLP;Evi5^{FRT}* animals. Red-dotted lines show the plasma membrane of PG cells. Dotted black boxes represent enlarged area to the right. Yellow arrowheads point at electron-dense vesicles found in PG cells of *PG >FLP;Evi5^{FRT}* larvae. **E** Close-up dark field high-voltage TEM photographs of electron-dense vesicles in PG cells isolated from *PG >FLP;Evi5^{FRT}* larvae. Dotted red boxes show electron dense vesicles, whereas green arrows depict residual bodies (lysosomes harboring undigested material). Scale bars are in micrometers (μm). **F** Box plots showing the average diameters (each sample contains 110 vesicles) and numbers of vesicles (three biological samples were analyzed) observed in the TEM sections of control and *PG >FLP;Evi5^{FRT}* genotypes. Asterisks: P-values based on a two-sided Student's t-test (***$p < 0.001$). **G** Lysotracker Blue staining of ring glands isolated from control and *PG >FLP;Evi5^{FRT}* animals. UV: ultraviolet light, BF: bright field. Scale bar = 250 μm. **H** Box plot showing lysosome abundance in ring glands based on Lysotracker Blue staining (normalized to RG area). $n = 5$ biological replicates. Asterisks denote P-values calculated with the two-sided Student's t-test (***$p < 0.001$). The whiskers in the box plots of 4C, 4F and 4H represent the range from the 5th to the 95th percentile. Source data are provided in the source data file.

in the control and experimental samples. This strategy yielded 75 proteins in the larval *Evi5^{FRT}* samples and 107 proteins for the S2 cell approach, which had 16 proteins in common, which is ~27 times higher than expected (Fig. 6A, Table 1, Supplementary Data 1 and Supplementary Data 2). In Table 1, we list the top 10 proteins of either approach, as well as any of the overlapping 16 proteins that are not in the top 10.

To mine the protein interaction data, we used term enrichment statistics and used the standard "biological process", "molecular function", "cellular component" terms, but we also searched for enrichment of proteins that interact with the proteins in the cohort (based on Biogrid), KEGG pathway, REACTOME, and protein domains (based on Interpro) (Supplementary Data 1). Our approach revealed several strongly and significantly enriched terms. REACTOME, for example, showed the top hit in the in vivo cohort as "Clathrin-mediated endocytosis", with a ~6.5-fold enrichment (9/107 proteins, $p = 5.5 \times 10^{-5}$). This finding was mirrored independently by BIOGRID, which revealed that the most commonly found interacting protein among proteins in the in vivo cohort was "Clathrin heavy chain" (20/107 proteins, $p = 4.5 \times 10^{-18}$). Our in vitro data showed no connection to Clathrin, suggesting that immortalized cells in culture, independent of an organism and adjacent tissues, rely less on this type of endocytosis. Other top-scoring enrichment terms were "vesicle-mediated transport", "cytoplasm", and "actin-filament-binding", all in various search categories (Supplementary Data 2). Taken together, *Drosophila* Evi5 appears to be mainly involved in the transport of Clathrin-coated vesicles, which has not yet been directly established in the vertebrate field.

Surprisingly, we found that Evi5 co-immunoprecipitated Fer1HCH in both approaches (Table 1). *Fer1HCH* encodes one of the two subunits of the ferritin nanocage, which is a 24-mer that stores up to 4500 iron atoms[64]. Most mammalian ferritin is cytoplasmic, but a small fraction is present in the serum, whereas *Drosophila* ferritin is secreted from cells and is thus predominantly present in the hemolymph[25,65]. An intriguing possibility is that fly ferritin is used to transport iron across the hemolymph to target tissues, but direct evidence is lacking[25,65]. The identification of Fer1HCH in two independent IP-MS experiments raised the possibility that Evi5 has roles in iron metabolism independent of vesicular transport of transferrins, possibly in the cellular uptake of ferritin from the hemolymph. We validated the interaction between Evi5 and Fer1HCH in S2 cells with Flag-tagged Evi5, using Rab11 as a positive and GFP as a negative control (Fig. 6B and Fig. S7). To further elucidate the link between Evi5 and ferritin, we crossed *PG>Evi5^{IR1}* to *Fer1HCH^{G188}*, a line that produces GFP-tagged Fer1HCH[24], to compare the GFP signal strength between Evi5-depleted and control PGs. Remarkably, *PG>Evi5^{IR1};Fer1HCH^{G188/+}* animals had ~4-fold higher Fer1HCH-GFP levels compared to controls (Fig. 6C, D). In contrast, the CNS showed roughly equal Fer1HCH-GFP levels between controls and *PG>Evi5^{IR1};Fer1HCH^{G188/+}* animals. To test whether this was caused by transcriptional upregulation of the *Fer1HCH-GFP* knock-in, we analyzed

the expression levels of the two endogenous *ferritin* genes in response to Evi5 depletion, which showed a minor increase (1.2–1.3-fold) of *Fer1HCH* and *Fer2LCH* (Fig. 6E). Given the minimal transcriptional upregulation, it was plausible that the 4-fold increase in Fer1HCH-GFP levels resulted from the failure to properly secrete cytoplasmic ferritin, resulting in its accumulation.

To explore the relationship between Evi5 and ferritin further, we carried out co-culture assays based on hemolymph from *Fer1HCH^{G188/+}* animals to test whether Fer1HCH-GFP would be endocytosed by cultured PGs that lack GFP (Fig. S5A). In brief, we failed to directly confirm the uptake of Fer1HCH-GFP into the PG, regardless of the conditions we used. Neither did we detect Fer1HCH-GFP in the PG nor did we detect GFP in Western blots of BRGCs. (Fig. S5A–C). There are several reasons as to why Fer1HCH uptake might be difficult to detect (see discussion), which is why we used indirect experiments instead.

Our approach was as follows: We reared donor larvae on iron-deficient and iron-replete diets to alter hemolymph ferritin levels, resulting in iron-deprived and iron-loaded ferritin/hemolymph samples, respectively[25,27]. Adding hemolymph isolated from *Fer1HCH^{G188/+}* larvae reared on an iron-rich diet substantially reduced the red autofluorescence in cultured *PG >AGBE^{IR}* ring glands, whereas using an iron-depleted diet instead did not reduce autofluorescence. In stark contrast, hemolymph addition could not rescue the autofluorescence associated with Evi5-depletion (Fig. S5A). Which hemolymph component rescued the autofluorescence of *PG >AGBE^{IR}* RGs? It was plausible that the rescue was caused by either ferritin or Tsf1 since hemolymph contains both proteins. To test this directly, we depleted Tsf1 and isolated hemolymph from *Tub >Tsf1^{IR}* as well as control larvae reared on an iron-rich diet since we had already demonstrated that the *Tsf1*-RNAi line is highly effective (Fig. 5F and Fig. S5D). Culturing BRGCs with Tsf1-depleted hemolymph rescued the autofluorescence of AGBE-depleted ring glands nearly as well as wild-type hemolymph (Fig. S5D), indicating that the rescue is due to a different iron carrier, which suggested that ferritin was the underlying reason for this rescue. To test whether Fer1HCH-GFP was in fact present in the hemolymph (i.e., the GFP moiety was not interfering with secretion), we conducted Western blotting of hemolymph and BRGC samples, which showed that Fer1HCH-GFP was indeed secreted (Fig. S5B & S5C).

In summary, while we were unable to provide evidence that Fer1HCH-GFP is taken up by PG cells, our data suggests that ferritin can rescue the iron deficiency associated with *AGBE*-RNAi. We conclude that cellular ferritin uptake serves as a source for iron. In the next section, we will provide additional evidence for this notion.

### Ferritin injection rescues *PG >FLP;Evi5^{FRT}* larvae

We developed a larval injection assay to determine whether providing exogenous holo-ferritin could rescue either developmental delays or the autofluorescence observed in *PG >FLP;Evi5^{FRT}* animals. We reasoned that injecting ferritin may increase iron levels in the PG and bypass the need for Evi5, provided that i) ferritin uptake is utilized by

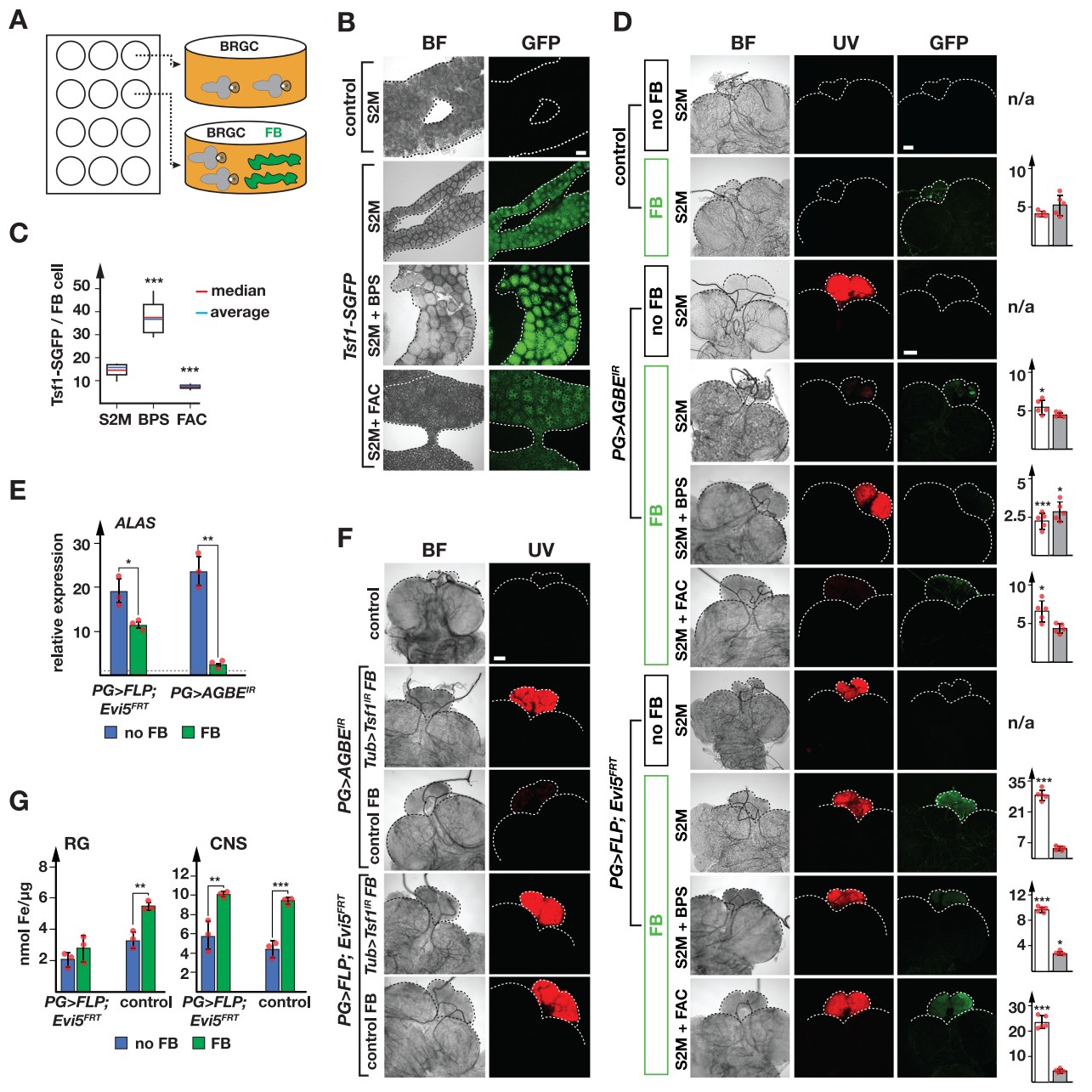

**Fig. 5 | Iron delivery via Tsf1 is dependent on Evi5. A** Illustration of co-culture assay. BRGCs were incubated in the presence or absence of fat bodies obtained from larvae with different genotypes, e.g., *Tsf1-GFP*-expressing larvae. Other variable parameters are larval diet, and compounds added to the Schneider 2 cell medium (S2M), such as iron-chelator ( + BPS) and iron ( + FAC). **B** Cultured tissues from control and *Tsf1-sGFP* larvae in S2M, S2M + BPS and S2M + FAC. BF: brightfield image. **C** Average Tsf1-sGFP signal (shown in **B**) per individual fat body cell (FB). Media were as follows: S2M, S2M + BPS and S2M + FAC. 10 cells were tested for each medium. The box plot whiskers represent the 5th to the 95th percentile range. **D** Images show brightfield (BF), protoporphyrin accumulation (UV) and Tsf1-sGFP uptake (GFP) of ring glands ($w^{1118}$ = control, *PG >FLP;Evi5^{FRT}* and *PG > AGBE^{IR}*, none harbour GFP transgenes). BRGC were co-incubated with fat bodies from *Tsf1-sGFP* larvae. Media as described in (**A**). Bar graphs represent Tsf1-sGFP signal in the RG (white) and the CNS (grey), normalized to area. Five biological replicates analyzed.

Scale bars = 150 μm. (no) FB: (no) fat bodies present. **E** Relative expression of *ALAS* in control BRGCs (dotted line: 100%), *PG >FLP;Evi5^{FRT}* and *PG > AGBE^{IR}* animals, after incubation with (green, FB) or without fat bodies (blue, no FB) from *Tsf1-SGFP* larvae. Fold changes calculated with ddCT method and error bars represent 95% confidence intervals, both are based on three biological replicates (each tested in triplicate). The center of error bars denotes the average. **F** Autofluorescence of cultured *PG >FLP;Evi5^{FRT}*, and *PG >AGBE^{IR}* RGs co-incubated with fat bodies from control and *αTub84B>Tsf1^{IR}* larvae. Scale bar = 150 μm. **G** Iron content in RG and CNS of control and *PG >FLP;Evi5^{FRT}* BRGCs cultured in the presence or absence of *Tsf1-sGFP* fat bodies (n = 5 biological replicates). Error bars in (**C**, **D**, **E**, **G**) are standard error, with the center indicating the average, and asterisks are P-values obtained via a two-sided, paired Student's t-test (***p < 0.001). **B**–**F** Larvae reared on standard food. Source data are provided in the source data file.

cells as a means to increase cellular iron concentrations and ii) Evi5 is not essential for intracellular transport of ferritin. To accomplish this, we injected control buffer (PBS), as well as holo- (iron-loaded) and apo- (iron-free) ferritin (in the form of equine spleen ferritin) into second

instar larvae (L2). As proof-of-principle, we used a genetic control, *PG>Fer1HCH^{IR}*, to test whether ferritin injection could rescue the lethality associated with ferritin depletion (Fig. 7A). Disrupting ferritin function in the PG resulted in 100% lethality, with most animals never

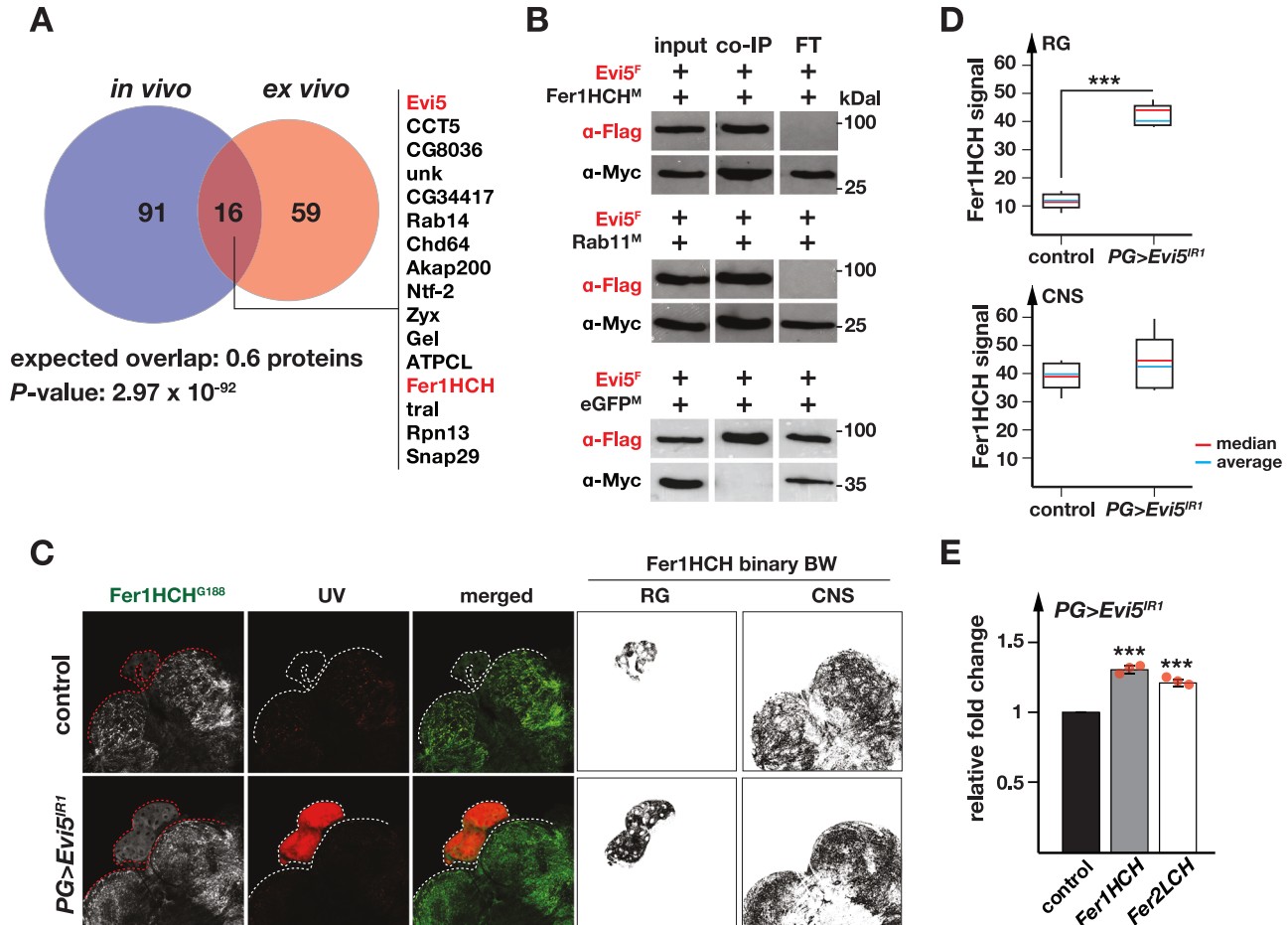

**Fig. 6 | Evi5 protein interactome. A** Venn diagram depicting MALDI-TOF results for proteins co-immunoprecipitated with Evi5. The in vivo assay was based on the Evi5^FRT line, which produces Flag-tagged Evi5. The ex vivo assay was carried out by transfecting S2 cells with plasmids encoding Flag-tagged Evi5. The Fer1HCH (red) was detected in both assays, along with 15 other proteins. The P-value is based on the one-sided Chi square test and shows the significance of the difference between the number of observed overlap proteins and the number of proteins that would be expected on average when two correspondingly sized lists of randomly picked *Drosophila* proteins are compared. **B** Co-immunoprecipitation assay of Evi5-3xFlag (Evi5^F), Fer1HCH-3xMyc (Fer1HCH^M), Rab11-3xMyc (Rab11^M) and eGFP-3xMyc (eGFP^M) expressed from corresponding plasmids transfected into S2 cells. In all experiments Flag-tagged Evi5 was used as bait (red). Myc-tagged Rab11 and eGFP were used as positive and negative controls, respectively. α-Flag and α-Myc stand for anti-Flag and anti-Myc antibodies. For uncropped images, see Fig S7A. **C** Ferritin-

GFP levels in BRGCs of *Fer1HCH^G188* (control) and *PG>Evi5^IR1;Fer1HCH^G188* animals. Fer1HCH-GFP image converted to grayscale on the left, but shown as green in the merged image. Black and white (BW) binary images generated to quantify Fer1HCH-GFP levels (in **D**). The red channel shows autofluorescence due to protoporphyrin accumulation. **D** Box plot showing Fer1HCH-GFP abundance in ring glands and the CNS of control and *PG>Evi5^IR1;Fer1HCH^G188* animals based on the binary BW images shown in (**C**). Red and blue lines indicate average and median values, respectively. Five biological samples tested. The whiskers in the box plots represent the range from the 5th to the 95th percentile. *** indicates $p < 0.001$, derived from the two-sided Student's t-test P-values. **E** qPCR analysis of *Fer1HCH* and *Fer2LCH* in RG samples of *PG>Evi5^IR1* and control animals. Fold changes calculated with ddCT method and error bars represent 95% confidence intervals, both are based on three biological replicates (each tested in triplicate). The center of error bars represents the mean. Source data are provided in the source data file.

reaching L3 and no animals reaching adulthood. Injecting equine holo-ferritin into L2 allowed *PG>Fer1HCH^IR* larvae to progress to L3 (50%), enter pupation (20%) and even reach adulthood (10%) (Fig. 7A). By contrast, injecting apo-ferritin accomplished no significant rescue. These data strongly suggested that the PG can endocytose holo-ferritin to either procure iron or to use the remaining capacity in ferritin to store excess iron for detoxification purposes.

Next, we tested whether injection of equine ferritin could rescue Evi5-loss-of-function animals. Whether L2 were left untreated or injected with PBS or apo-ferritin, *PG>FLP;Evi5^FRT* populations all showed ~5-day delays. By contrast, injection of non-protein-bound iron (in the form of FAC) and heat-denatured holo-ferritin resulted in a ~24 h improvement. The idea of denaturing ferritin[66] was that the injected sample has exactly the same chemical composition as holo-ferritin (i.e., protein + iron) but should not be taken up by cells due to the disrupted ferritin nanocage structure. However, it was possible

that injected ferritin was disassembled in the hemolymph, resulting in the release of elemental iron that was taken up by cells. In this case, FAC, apo- and holo-ferritin should all have comparable results. Importantly, this was not the case. *PG >FLP;Evi5^FRT* L2 injected with holo-ferritin showed clearly the most significant rescue, with a nearly 3-day faster development than injection with apo-ferritin (Fig. 7B). These data demonstrated that holo-ferritin must be intact in order to rescue Evi5-loss-of-function animals.

Next, we dissected BRGC from injected larvae to examine the amount of autofluorescence in the PG (Fig. 7C). While one can see differences in the autofluorescence levels, the results were not obvious without quantification. We noticed that injection of holo-ferritin reduced overall autofluorescence to a higher degree than all other compounds but it did not eliminate autofluorescence altogether. We then analyzed *Alas* expression to quantify the effect of the injected compound on heme biosynthesis (Fig. 7D). Using this strategy, we

**Table 1 | List of proteins detected with MALDI-TOF as a result of co- co-immunoprecipitation with Evi5**

| Symbol[A] | Description[B] | *in vivo* assay | | *ex vivo* assay | |
|---|---|---|---|---|---|
| | | rank (/ 107)[C] | IP-MS Score[D] | rank (/ 75)[C] | IP-MS Score[D] |
| Cindr | Endocytosis/border cell migration /actin filament organization | 1 | 222.67 | nd | nd |
| Pzg | Ecdysone signaling/ cell cycle regulation/ chromatin organization | 2 | 206.9 | nd | nd |
| Evi5 | GTPase activator activity/ border follicle cell migration | 3 | 128.3 | 2 | 232 |
| Chc | Endocytosis/ Clathrin coat assembly/ autophagy regulation | 4 | 71.87 | nd | nd |
| CCT5 | Protein folding | 5 | 60.96 | 9 | 28.19 |
| CG8036 | - | 6 | 58.98 | 7 | 48.26 |
| Hsc70-4 | protein folding/ selective autophagy regulation/ vesicle transport | 7 | 57.56 | nd | nd |
| unk | Neuron differentiation/ compound eye development | 8 | 40.66 | 15 | 19.44 |
| Dap160 | Vesicular endocytosis and transport | 9 | 38.64 | nd | nd |
| CG34417 | Actin cytoskeleton organization | 10 | 38.33 | 8 | 42.22 |
| Zip | Mitotic cytokinesis/ border follicle cell migration | nd | nd | 1 | 1037 |
| β-Tub85D | Cell cycle regulation/ microtubule cytoskeleton organization | nd | nd | 3 | 186.26 |
| Jar | Endocytosis/border cell migration /actin cytoskeleton organization | nd | nd | 4 | 82.56 |
| Hsc70-1 | protein folding/ vesicle transport | nd | nd | 5 | 78.32 |
| CG1737 | - | nd | nd | 6 | 48.6 |
| Akap200 | Actin cytoskeleton organization/ protein localization to membrane | 31 | 14.04 | 10 | 24.02 |
| Rab14 | Endocytic recycling/ phagolysosome assembly | 20 | 21.55 | 18 | 15.12 |
| Chd64 | Muscle contraction/ juvenile hormone signaling | 21 | 21.21 | 22 | 12.60 |
| Ntf-2 | Protein import into nucleus | 37 | 11.38 | 51 | 5.6 |
| Zyx | Tracheal system regulation/ cell adhesion/ hippo signaling | 40 | 10.48 | 25 | 11.39 |
| Gel | Actin filament polymerization | 48 | 9.65 | 19 | 14.83 |
| ATPCL | Acetyl-CoA biosynthesis/ fatty acid biosynthesis | 50 | 9.07 | 55 | 5.26 |
| Fer1HCH | Iron homeostasis | 53 | 8.17 | 38 | 7.58 |
| tral | Actin cytoskeleton organization/ ER organization | 57 | 7.69 | 23 | 11.99 |
| Rpn13 | Ubiquitin-dependent protein catalysis | 58 | 7.31 | 16 | 16.53 |
| Snap29 | Endocytosis/ vesicle fusion/ Golgi organization /autophagosome maturation | 100 | 2.69 | 69 | 4.3 |
| Rab11* | Endocytosis and exocytosis regulation/ autophagosome maturation | - | 24.09 | - | 11.45 |

[A] Green font represent proteins involved in selective autophagy, and red font refers to proteins that function in iron homeostasis.
[B] Descriptions summarizing protein function were obtained from FlyBase.
[C] Ranks represent protein number based on IP-MS score in in vivo (total:107) and ex vivo (total:75) assays.
[D] IP-MS score is the sum of the ion scores of all identified peptides in the MALDI-TOF mass spectrometry.
*Rab11 was detected in both assays, however, the protein was also detected in controls, and was therefore not included in the overlap of 16 proteins (see A). nd = not detected.

observed the strongest reduction in *Alas* expression occurred after holo-ferritin injection (a 3-fold reduction vs. PBS injection and a 2-fold reduction vs. denatured holo-ferritin). Interestingly, the amount of injected iron via FAC was ~ 11,000-fold higher than the iron we injected via holo-ferritin, assuming the best case scenario that ferritin is fully loaded with iron. Despite this difference, the effects of denatured holo-ferritin closely resembled those observed with elemental iron injection (Fig. 7B–D). We conclude that these results provide strong evidence that ferritin is a key iron resource and is endocytosed by cells. It appears that the ferritin uptake pathway is partially functional in loss-of-Evi5-function animals, which is consistent with the notion that Rab proteins have intrinsic GTPase activity, and as such, some vesicular trafficking will occur even in the absence of Evi5.

## Discussion

In this report, we have shed light on many aspects of *Drosophila* iron trafficking, pointing to the fact that Evi5, Rab5, Rab11, Tsf1 and ferritin act together to ensure proper intracellular iron transport in an iron-rich tissue, the *Drosophila* PG. While Evi5 has never been directly linked to cellular iron biology, its interaction with Rab11 – a protein involved in the recycling of transferrin proteins, does provide an indirect link[49,53]. Mutations in Rab11 have been shown to block transferrin recycling and trigger the degradation of transferrin[67]. We have shown that blocking Evi5 function in a PG-specific manner impaired (but didn't completely block) vesicular iron trafficking and resulted in an increased vesicle abundance. Our co-culture assays demonstrated that fat body-derived Tsf1 was used to import iron into the PG and that this was sufficient to rescue the iron deficiency phenotypes of *PG >AGBE*[IR] RGs. However, when we used *PG >FLP;Evi5*[FRT] RGs instead, no rescue was seen, as Tsf1 failed to prevent protoporphyrin accumulation despite being endocytosed at high levels into PG cells (Fig. 5D). Clearly,

high cellular levels of Tsf1-GFP do not equate high iron availability, likely because vesicular iron transport slows down dramatically when Evi5 is impaired. PG-specific Evi5 impairment was not lethal, likely because Rab11 has intrinsic GTPases activity, albeit at much lower levels in the absence of Evi5. Indeed, our synchrotron analysis demonstrated (Fig. 2F) that loss-of-Evi5-function decreased intracellular iron levels in the PG, and the less sensitive ferrozine assay showed the same trend (Fig. 5G). Consistent with this, *Evi5* mutant phenotypes were rescued significantly with hemin administration and partially with iron feeding, which further supported the notion of impaired intracellular iron trafficking (Fig. 2A–E).

The finding that Evi5 physically interacts with ferritin was surprising and opens new avenues for investigating Evi5's roles in iron homeostasis. It has been proposed that ferritin may be used for systemic iron delivery in insects, which may be particularly critical for tissues with high iron demands[10,65]. Alternatively, it is possible that ferritin retrieval is relevant only under iron-deprived conditions. More experiments are needed to elucidate the role of extracellular ferritin. Our data indicate that Evi5 has a key role in trafficking iron via endocytosed transferrin and possibly in the release of iron from ferritin as well. We showed that larval injections of exogenous holo-ferritin, but not apo-ferritin, moderately reduced protoporphyrin accumulation but substantially shortened the developmental delays of *PG >FLP;Evi5*[FRT] animals (Fig. 7B–D). Denatured holo-ferritin showed intermediate results, but was clearly less effective than holo-ferritin (Fig. 7B–D), strongly suggesting that cells endocytose intact holo-ferritin to retrieve iron from the hemolymph. Denatured ferritin may be still partially effective because heat denaturation of ferritin is usually incomplete, as was previously reported[66]. In agreement with this idea, we also observed elevated GFP levels in the PG of *PG>Evi5*[IR1];*Fer1HCH*[G188/+] animals (Fig. 6C, D), which may have been

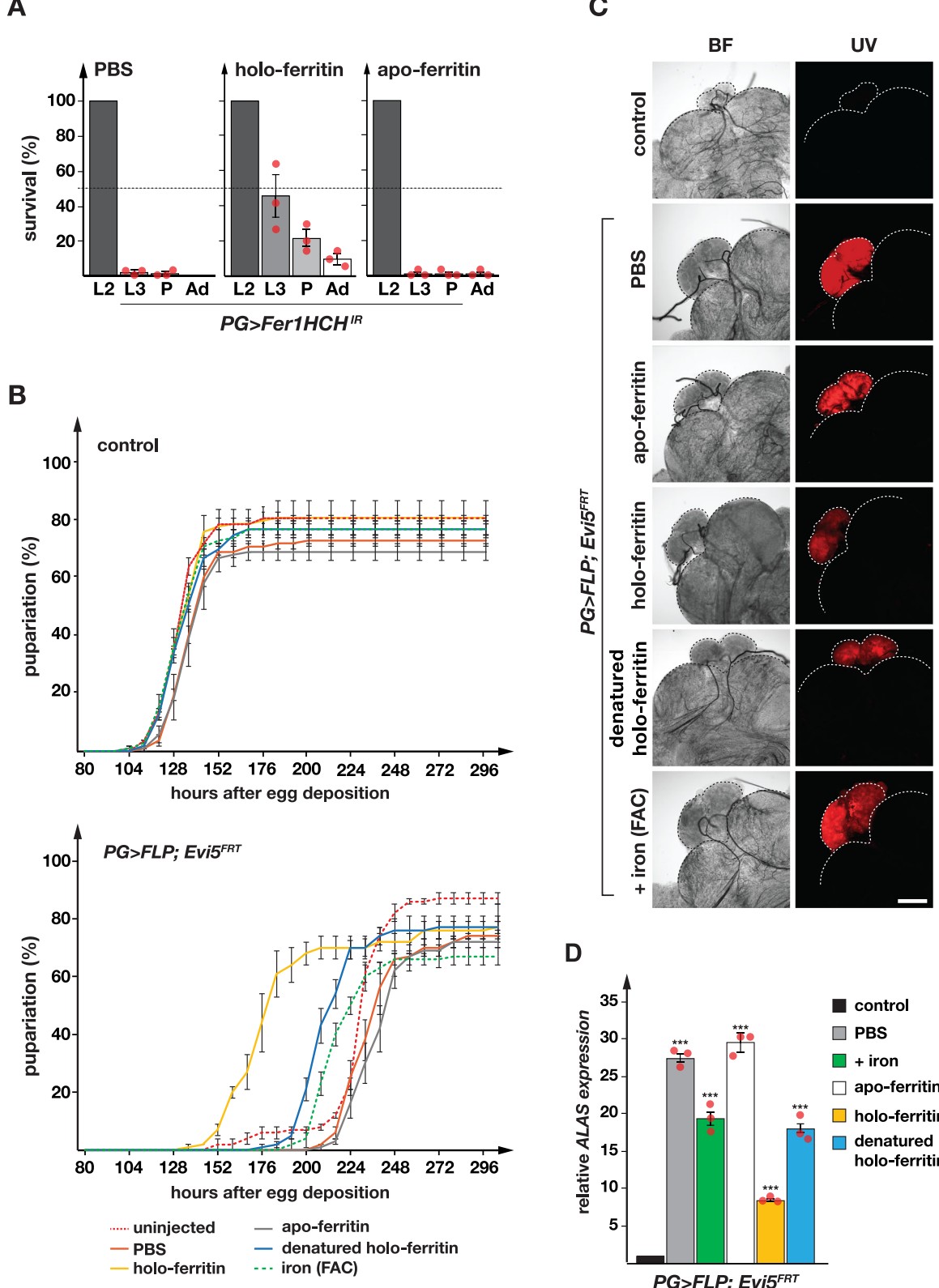

**Fig. 7 | Injection of ferritin partially rescues Evi5-impaired larvae. A** Survival analysis of *PG>Fer1HCH[IR]* animals injected with ferritin solutions. Commercially obtained apo- and holo-ferritin isolated from equine spleen were injected into of *PG>Fer1HCH[IR]* second instar larvae and scored for surviving animals at larval, pupal and adult stages. L2: 2nd instar, L3: 3rd instar, P: pupae, Ad: adults. **B** Charts depicting the elapsed time for control (*PG > w[1118]*) and *PG >FLP;Evi5[FRT]* embryos to reach the prepupal stage. Denatured holo-ferritin served as a negative control. In (**A**, **B**) error bars represent standard error of three biological replicates (each = 50 individuals), with the center indicating the average. **C** PG cells protoporphyrin accumulation analysis of *PG >FLP;Evi5[FRT]* larvae injected with indicated compounds under UV light. BF: bright field. Scale bars = 250 μm. **D** Expression analysis of the *ALAS* in ring gland samples isolated from injected *PG >FLP;Evi5[FRT]* larvae. Fold changes calculated with ddCT method and error bars represent 95% confidence intervals, both are based on three biological replicates (each tested in triplicate). The center of error bars denotes the average. Asterisks denote *P*-values calculated with the two-sided Student's t-test (***$p < 0.001$). Source data are provided in the source data file.

caused by the uptake of Fer1HCH-GFP from the hemolymph coupled with impaired release. However, even though transcriptional upregulation of the two ferritin genes was a mere 1.25-fold (Fig. 6E), we cannot rule out that increased transcript levels caused the 4-fold increase in Fer1HCH-GFP (Fig. 6D) in combination with transcript stabilization or protein perdurance.

We were unable to demonstrate direct uptake of Fer1HCH-GFP into the PG in our co-culture experiments (Fig. S5). This may have been for several reasons. The first possibility is that Fer1HCH-GFP-containing ferritin nanocages are functionally constrained and not properly endocytosed. It is clear that Fer1HCH-GFP-containing ferritin nanocages are excreted from cells and thus present in the hemolymph (Fig. S5B) and capable of storing iron[64]. Since GFP is expected to be on the outside of the ferritin shell[64], the GFP moiety may interfere with a putative ferritin receptor. Alternatively, it is possible that the GFP:iron ratio is simply too low to result in detectable fluorescence levels. A fully iron-loaded ferritin nanocage is estimated to harbor ~4,500 iron atoms. The theoretical maximum of Fer1HCH-GFP moieties for a single ferritin shell is 12, but we expect this number to be considerably lower, since homozygous $Fer1HCH^{G188/G188}$ animals are not viable[24], indicating that pure Fer1HCH-GFP/Fer2LCH shells are not functional or do not form for steric reasons. Our experiments were carried out with $Fer1HCH^{G188/+}$ heterozygotes, and as such, we estimate the GFP:iron ratio to be ~1:1000 for iron found in iron-loaded Fer1HCH-GFP-harbouring ferritin nanocages. This ratio may be even lower if GFP-containing ferritin is competing with wild-type ferritin during endocytosis or assembly (cells will contain a range of ferritin shells ranging from zero to a few GFP moieties). In contrast, _Drosophila_ transferrin binds iron with a 1:1 stoichiometry[68], resulting in a GFP:iron ratio of ~1:1 for Tsf1-GFP-producing flies. As such, the uptake of iron via ferritin in our co-culture studies may result in a 1000-fold lower GFP signal compared to the Tsf1-GFP experiment and thus evade detection in our hands.

The co-culture experiments do, however, provide strong indirect evidence that ferritin endocytosis occurs in the PG. Specifically, we show that hemolymph taken from iron-fed $Fer1HCH^{G188/+}$ larvae substantially reduces the autofluorescence phenotype of $PG>AGBE^{IR}$ RGs, but not that of $PG>FLP;Evi5^{FRT}$ RGs (Fig. S5A). This rescue could be due to the presence of Tsf1, GFP-ferritin or wild-type ferritin in the added hemolymph. Adding elemental iron in the form of FAC neither rescues the autofluorescence of $PG>AGBE^{IR}$ nor $PG>FLP;Evi5^{FRT}$ RGs. To address whether the rescue was due to Tsf1, we showed that Tsf1-depleted, iron-fed larvae dramatically rescued the autofluorescence of $PG>AGBE^{IR}$ ring glands, but not that of $PG>FLP;Evi5^{FRT}$ ring glands (Fig. S5D). Taken together, our data, while not providing direct evidence for ferritin uptake into PG cells, strongly suggests that ferritin is endocytosed and used as a source of iron.

A not fully understood aspect of iron metabolism is the release of iron from holo-ferritin. Ferritin iron recovery, also known as ferritinophagy, is a selective autophagic process that recovers iron from ferritin nanocages. In mammalian cells, ferritin enters lysosomes via binding to NCOA4, a nuclear receptor co-activator that double-functions as a lysosomal receptor[69]. Studies in mice demonstrated that mutations in NCOA4 impair systemic iron homeostasis and induce anemia[70]. This suggests that ferritinophagy can occur in normal and iron-depleted conditions. Intriguingly, intermediate structures formed during ferritin autophagy are electron-dense due to the presence of undigested or completely digested holo-ferritin (siderosomes and hemosiderin, respectively). TEM microscopy reveals siderosomes and hemosiderin as electron-dense and iron-rich lysosome-based structures. We also observed a high degree of electron-dense vesicles in the PG cells of control and $Evi5^{FRT}$ animals. While these dense structures are highly uniform in the control, they are abnormally dispersed in the cytosol of mutant cells and have variable densities (Fig. 4C). Using TEM, we showed that the electron-dense vesicles of $Evi5^{FRT}$ animals

resemble lysosomes in size and shape, and likely form residual bodies that originate from autophagosomes and secondary lysosomes (Fig. 4D).

Ferritin autophagy has not been described in _Drosophila_ yet[71]. What could be the biological context that requires ferritinophagy? During larval stages, ecdysone is produced in pulses prior to larval molts and puparium formation[20,72,73], which equates a periodically high demand for iron[16] (Fig. S6A). Ferritinophagy releases a large amount of iron into cells[14]. Thus, we hypothesize that Evi5 directs cytosolic or endocytosed holo-ferritin to lysosomes to replenish cellular iron levels prior to an upcoming ecdysone peak (Fig. S6A). The interaction of Evi5 with Chc, Hsc70-4, Rab11 and Fer1HCH provide support for this hypothesis (Table 1). Chc encodes Clathrin heavy chain, which is crucial for autophagosome initiation and lysosome formation, while endosomal co-chaperone Hsc70-4 contributes to generating multivesicular bodies targeted to lysosomes[74,75]. Consistent with this, Rab11, an established Evi5 target, governs endosomal trafficking in both autophagy- and recycling-related processes[76]. In addition, $PG>Fer1HCH^{IR}$ and $PG>Hsc70-4^{IR}$ animals arrest development at the L2 stage and $PG>Chc^{IR}$ animals display developmental delays, similar to the severe effects seen upon _Evi5_ and _Rab11_ depletion in the PG (Fig. S6B). Further support stems from the PG-specific loss of _Atg8a_ function, a key player of selective autophagy, which also caused larval arrest (Fig. S6B). Interestingly, the _Drosophila_ proteins Atg8a and Fer1HCH genetically interact, and the human Atg8 ortholog GABARAP physically interacts with the ferritin lysosomal receptor NCOA4[71]. Taken together, we believe it is conceivable that Evi5 facilitates the release of iron from holo-ferritin via ferritinophagy, which we plan to address in future studies.

Finally, Evi5 has been identified in multiple GWAS studies as a high-risk locus for MS[35–39]. Multiple sclerosis is a neurodegenerative disease that leads to oligodendrocyte destruction, demyelination, remyelination and astrocytic scar formation[77]. Aberrant iron deposition in the grey and white matter of MS patients is a hallmark of the disease that further worsens demyelination and immune responses at lesion sites[78–80]. MS patients have elevated ferritin levels (hyperferritinemia) in the cerebrospinal fluid and display iron accumulation around the plaques[81,82]. MRI imaging of grey matter from MS patients has linked brain shrinkage, disability progression, and cognitive impairment with iron accumulation in the brain[78]. While MRI imaging techniques and histochemical studies demonstrated iron accumulation in the form of ferritin and hemosiderin in MS plaques[40,78], the exact molecular mechanism of iron deposition remains unknown.

Ferric iron bound to ferritin is the most abundant trace metal in a healthy brain[83]. In MS patients, the destruction of oligodendrocytes and myelin at the sites of plaque formation causes heme leakage followed by activation of macrophages and microglia cells to absorb the released heme/iron[83–85]. Activation of microglia releases proinflammatory cytokines that perturb the expression of iron transporters like DMT1 and ferroportin. Furthermore, the release of hydroxyl and nitric oxide radicals as well as ROS formation due to iron release, cause the upregulation of _DMT1_, _Tfr1_, _Fpt1_, which increases cytosolic ferritin levels[84]. Microarray analysis of MS lesions demonstrated that expression of ferritin genes, _Tfr1_, _DMT1_ and _Zip14_ genes are all elevated at sites of inflammation[1]. As such, MS etiology affects key players of iron transport and iron homeostasis, and assessing iron levels in patients has been proposed to serve as a marker for the severity of the disease. Intriguingly, iron-loaded ovoid structures around the lesions of myelinated white matter in MS patients resemble the engulfed vesicles we observed in the Evi5 mutant PG cells. Therefore, this study opens up the fascinating possibility that the reason Evi5 has been linked to MS lies in its role in cellular iron trafficking and that MS - at its core - is caused or exacerbated by a disruption of cellular iron transport.

## Methods

### *Drosophila* stocks and husbandry

The fly stocks used in this study are listed in Table S1 and were obtained from the Bloomington *Drosophila* Stock Center and Vienna *Drosophila* Resource Center. We used CRISPR/Cas9 to generate an *Evi5$^{FRT}$* knock-in allele (Fig. S1A). We also generated three transgenic lines using the PhiC31 system, namely *UAS-Tsf1-3XFlag*, *UAS-Tsf2-3XFlag* and *UAS-Tsf3-3XFlag*. *Phm22-Gal4* and *Fer1HCH$^{G188}$* were kind gifts from Dr. Michael O'Connor's and Dr. Fanis Missirlis's labs, respectively. Stocks were maintained on a standard cornmeal diet, but Nutri-Fly food (Genesee Scientific, #66113) was used as a base medium when we modified the composition of the diet.

### Survival and RNAi screening studies

Prior to performing survival and developmental progression studies, we reared all flies on Nutri-Fly media for at least one generation. All RNAi screening procedures were based on the standard formulation of Nutri-Fly (Bloomington formula, Genesee Scientific), but developmental progression experiments for *Evi5*-RNAi and *Evi5$^{FRT}$* lines were carried out both with standard as well as modified Nutri-Fly media. Modified Nutri-Fly media were prepared by adding specific compounds with the following concentrations: 1 mM FAC (Ferric Ammonium citrate, Sigma #F5879), 120 μM BPS (Bathophenanthrolinedisulfonic acid, Sigma #146617), 100 μM hemin (Sigma #H9039), 1 mM ZnPP (zinc protoporphyrin IX, Sigma #691550) and 120 μg/ml 20E (20-hydroxyecdysone, Cedarlane #H918750) supplemented media. Developmental progression was evaluated by using 3 × 50 eggs per genotype in 8 h intervals from the embryonic to pupal stages. All experiments were carried out at 25 °C in a chamber with 60–70% humidity.

### Construction of CRISPR/Cas9 FRT and transgenic lines

We used CRISPR/Cas9 homology-directed repair to generate an excisable knock-in allele of Evi5 by replacing the endogenous allele with an FRT-flanked version (*Evi5$^{FRT}$*) (Fig. S1A). To design gRNAs, we obtained target regions from the FlyBase database which were subsequently submitted to the CRISPR Optimal Target finder (http://targetfinder.flycrispr.neuro.brown.edu). The selected gRNAs were then validated by Sanger sequencing of the corresponding genomic loci of the Vas.-Cas9 line (Bloomington #51323), which we used for embryo injections. All donor template fragments were further amplified from the genomic DNA of the Vas.Cas9 line (Bloomington #51323) via PCR and confirmed by sequencing. We then cloned the gRNAs into the pCFD5 plasmid (Addgene, #73914), whereas donor templates were inserted into the pDsRed-attP vector (Addgene #51019).

For transgenic lines, full-length cDNAs of *Evi5* (#GH14362), *Tsf1* (#LP08340), *Tsf2* (LD22449) and *Tsf3* (FIO3676) were obtained from *Drosophila* Genomic Resource Center. The Evi5 wild-type cDNA was cloned into the pBID-UASC-GV plasmid (Addgene #35204), a PhiC31 vector. This adds an mVenus-tag to the C-terminus of Evi5 (UAS-Evi5$^{WT}$-mVenus). The *Evi5* Arg160 to Ala (*Evi5$^{R160A}$*) mutant cDNA was generated from the UAS-Evi5$^{WT}$-mVenus plasmid via PCR mutagenesis. For transferrin transgenic lines, we added sequences encoding C-terminal 3XFlag tags to all cDNAs and cloned them into pBID-UASC-FG (Addgene #35201), also a PhiC31 vector. All fragments used for cloning were amplified with Q5 High-Fidelity DNA Polymerase, and they were fused together via the Gibson assembly master mix (NEB, #E2611). Primers used in this study are listed in Table S2.

### Embryo injections

For embryo injections, we used standard procedures[86] with approximately 300–500 embryos injection per construct. The following concentrations were used for plasmids. We used 150 ng/μl for double gRNA vectors (pCFD5) and 500 ng/μl for the CRISPR/Cas9 donor template vector (pDsRed-attP) to generate the *Evi5$^{FRT}$* allele. For

PhiC31-based plasmids, we used 500 ng/μl to produce the transgenic transferrin lines. The CRISPR/Cas9 injection was performed at the University of Alberta, and the transgenic vectors were injected by GenetiVision.

### Larval injections

Our larval injection protocol was adapted from a procedure developed for *Tribolium* larvae[87]. We collected 50 early L2 (24-26 hr after egg deposition) of *PG > FLP;Evi5$^{FRT}$*, *PG>Fer1HCH$^{IR}$* and control animals. Larvae were washed 3X in PBS and dried immediately on a paper towel. Larvae were aligned on double-sided sticky tape that had been glued to a glass slide. The larvae were covered with halocarbon oil (Sigma, #H8898), and liquids were injected into the T3 or A1 segments of the larvae. The following compounds/concentrations were injected: 250 μM FAC, 5 pM holo-Ferritin (Equine spleen, Sigma, #F4503), 5 pM apo-ferritin (Equine spleen, Sigma, #A3660) and 5 pM denatured holo-ferritin. Denatured holo-ferritin was generated via boiling holo-ferritin (Equine spleen, Sigma, #F4503) for 30 min at 95 °C. The holo-Ferritin and apo-ferritin were diluted in PBS buffer.

All injections were carried out with the microinjector apparatus (Tritech Research) at 5 PSI and 0.2 s intervals. Injection volume was 1 μl.

### Transmission Electron Microscopy (TEM)

For TEM analysis, we used the brain-ring gland complexes (BRGC) isolated from third instar larvae (L3) staged at 40–44 h after the molt from second instar larvae (L2) to L3. The BRGCs were dissected in PBS buffer and fixed overnight in the TEM fixative buffer (2.5% glutaraldehyde, 2% paraformaldehyde and 1X PBS). The post-fixation, serial ethanol dehydrations, super resin mold formation, cross-sectioning and uranyl acetate-stained girds preparations were performed at the microscopy facility at the University of Alberta (Microscopy Unit; Room CW225; Biological Sciences Bldg.). All images were taken with a Transmission Electron Microscope at 80 kV accelerating voltage (Philips FEI, Model = Morgagni 268).

### High-voltage transmission electron microscopy

We used uranyl acetate-stained girds for High-voltage TEM analysis by coating the girds with a layer of carbon. Both sides of the girds were coated with 5 nm carbon by the Leica ACE 600 coater. The thickness of the coating was monitored by a quartz microbalance. The High-voltage TEM images were taken with a JEOL JEM-ARM200CF S/TEM electron microscope at an accelerating voltage of 200 kV. All images were taken at the microscopy facility at the University of Alberta (Microscopy Unit; Room W1-060 ECERF; nanoFAB Bldg.).

### Synchrotron iron analysis

The BRGCs isolated from 36, 40 and 44 h control L3 and 36, 40, 44, 60, and 74 h *Evi5$^{FRT}$* L3 were dissected in 0.25 M sucrose buffer. Dissected BRGCs were placed on a Thermanox Cover Slip (Fisher Scientific # 50949476) and air-dried. X-ray fluorescence (XRF) iron maps were generated at Stanford Synchrotron Radiation light source (SSRL) at the SLAC National Accelerator Laboratory (https://www6.slac.stanford.edu). The images were analyzed by the SMAK Microprobe Analysis Toolkit[88].

### Quantitative real-time PCR (qPCR)

Ring glands (RGs) and BRGCs were dissected from 40 to 44 h old L3 and divided into three biological replicates. RNA samples were extracted with the Qiagen RNeasy extraction kit and reverse-transcribed by Applied Biosystems™ High-Capacity cDNA Reverse Transcription kit (Thermo Fisher Scientific, #4374967). Synthesized cDNAs were then diluted 1:20 for 1 μg total RNA per reaction. Then NEB SYBR green Luna® Universal qPCR master mix (NEB, #M3003L) was used for qPCR by QuantStudio 6 Flex qPCR machine (Applied Biosystems). The ΔΔCT method was used to calculate fold changes, and

transcript levels were normalized to *rp49* expression. Statistical significance was tested with the Student's t-test, and error bars represent 95% confidence intervals. qPCR Primers are listed in Table S2.

## Ex vivo constructs and transfection

We used *Drosophila* S2 cells for our ex vivo experiments. Cells were cultured in Schneider insect medium (Sigma, Lot#RNBH8523) with 10% heat-inactivated fetal bovine serum (FBS) and 1% streptomycin-penicillin. For co-localization assays, the sequence corresponding to the C-terminus of the full-length Evi5 cDNA was ligated to a fragment encoding mCherry. For Tsf1, Tsf2 and Tsf3 cDNAs, we first identified their predicted secretion signal peptides via SignalP 5.0[89] and then removed the corresponding sequences in the subsequent cloning steps. The modified cDNAs were then tagged with eGFP C-terminally (Tsf1$^{\Delta SP}$-eGFP, Tsf2$^{\Delta SP}$-eGFP and Tsf3$^{\Delta SP}$-eGFP) and cloned into the pAFW plasmid (without the Flag sequence). The eGFP protein was further cloned separately in the pAFW plasmid and used as the control in the colocalization assay.

For co-immunoprecipitation (co-IP) experiments, cDNAs were modified so that Evi5 was tagged with 3XFlag, and Fer1HCH, Rab11 and eGFP were tagged with 4XMyc epitopes, all at the C-terminus. The tagged cDNAs were then cloned into the pAFW (Evi5) or pAMW (Fer1HCH, Rab11 and eGFP) plasmids, respectively.

All fragments were amplified with Q5 High-Fidelity DNA Polymerase and fused via the Gibson assembly master mix (NEB, #E2611). Primers are listed in Table S2.

## Tissue co-culture assays

Ex vivo culturing was done according to a method described before in ref. 90; however, we did not add insulin and 20E to the media. BRGCs and fat bodies were collected from L3 40–44 h after the L2/L3 molt. We used Schneider insect medium with 10% heat-inactivated FBS and 1% streptomycin-penicillin for tissue culture. Prior to dissection, all L3 were washed 3X in PBS buffer and all materials (forceps, dissecting plates, and microscopes) were sterilized using 70% ethanol. 15–20 BRGCs and fat bodies were quickly dissected in Schneider medium and transferred into 24-well tissue culture plates (FALCON #353047). For co-culturing, we placed 15–20 BRGC and 15–20 larval fat bodies into each well. Cultured BRGCs and fat bodies were maintained in a standard 25 °C incubator for 24 h. We added 1 ml media into each well and changed the media every 12 h. Supplemented Schneider media were made by adding: 1 mM FAC (Sigma #F5879), 500 μM BPS (Sigma #146617). Incubated samples were mounted in a standard mounting medium, and all images were taken with A Nikon C2si Confocal Microscope at the University of Alberta.

## Ferrozine Assay

The Ferrozine assay was carried out as described previously[27,91]. We cultured BRGCs and fat bodies in Schneider insect medium as described in the section above. ~75 RGs and 15 CNS were then dissected and transferred into 65 μl of lysis buffer (50 mM Tris-HCL, pH 7.4, 150 mM NaCl, 1 mM EDTA, 0.1% Triton X-100 and 0.1% glycerol) separately. Tissues were homogenized, and total proteins were extracted. Protein concentrations were measured by QubitTM Protein assay (Invitrogen, #Q33212). We then added 38.5 μl of cell lysates to 8.5 μl of concentrated HCl, and boiled samples at 95 °C for 20 min. 25 μl of boiled samples were added to 10 μl of 75 mM ascorbate (Sigma-Aldrich, #A92902-100G), followed by vortexing and centrifugation of the samples at 16,000 g. 10 μl of 10 mM ferrozine (Sigma-Aldrich, #160601-1 G) was added to each sample, the samples vortexed and centrifuged. Finally, we added 10 μl of 5 M ammonium acetate to each tube and vortexed samples. Samples were added to a 96-well flat bottom UV-star plate (Sigma-Aldrich, # M3812-40EA), and a plate reader (VICTOR Nivo Multimode Microplate Reader) was used to measure absorbance at 562 nm. Iron concentrations were calculated

according to the following formula:

$$Fe\,(pmol/\mu l) = ((DOD_{562} \times 47/38.5 \times 65/25)\,/\,27,900) \times 10^6$$

## Larval hemolymph collection

Hemolymph extraction from larvae was based on a protocol published by J. Bag and M. Mishra[92]. Samples were collected from 100 L3 (40–44 h) in batches of 15–20 animals. Larvae were washed 1X PBS and pierced gently with forceps (#5) near the larval mouth hooks. The hemolymph samples were immediately collected and snap-frozen by liquid nitrogen to minimize melanization. To analyze extracted hemolymph on Western blots, 10 μl 5X Laemmli protein loading buffer was added to 40 μl hemolymph. Next, samples were boiled at 95 °C for 5 min and subjected to SDS-PAGE. For ex vivo culture, 100 μl hemolymph was immediately mixed with 1 ml Schneider insect medium to avoid coagulation. The hemolymph-supplemented medium was then used for culturing BRGCs.

## MALDI-TOF mass spectrometry of whole larvae and S2 cells

Mass spectrometry assays based on whole larvae were essentially carried out as described before[16]. We collected 150-200 L3 staged larvae at 40–44 h and washed them in PBS. Samples were then washed in PBST (1X PBS with 0.1% Triton X-100) and treated with fixation buffer (1X PBST with 0.2% Formaldehyde). Reactions were terminated with a quenching solution (0.25 M glycine in 1X PBST). Total proteins were then extracted by using 1X lysis buffer (25 mM Na-HEPES pH 7.5, 75 mM NaCl, 0.5 mM EDTA, 10% glycerol, 0.1% Triton X-100, proteinase inhibitor cocktail).

For the co-immunoprecipitation assays based on cultured cells, we used the *Drosophila* S2 cells. The cells were transfected with pAFW-Evi5-3XFlag via the Calcium-Phosphate transfection method (Thermo Fisher Scientific, #K278001). Transfected cells were then harvested and 3X washed with PBS, followed by treatment with 1X lysis buffer (50 mM Tris-HCL, pH 7.4, 150 mM NaCl, 1 mM EDTA, 0.1% NP-40 and 1x proteinase K inhibitor).

Each individual assay was based on a single sample. Protein levels were normalized based on protein concentrations obtained with the QubitTM Protein assay (Invitrogen, #Q33212). For the pull-down assays, M2 Flag beads (Sigma-Aldrich, #A2220) were added to Chromotek sct-50 spin columns and incubated with sample lysates for 4 h at 4 °C following instructions of the manufacturer. Beads were rinsed with wash buffer 1 (25 mM Na-HEPES pH 7.5, 75 mM NaCl, 0.5 mM EDTA, 10% Glycerol, 0.1% Triton X-100) and wash buffer 2 (25 mM Na-HEPES, pH 7.5, 75 mM NaCl, 0.5 mM EDTA, 10% glycerol). Pulled-down proteins were eluted and separated on a 12% SDS gel. Proteins were visualized with Coomassie Brilliant Blue and SDS gel pieces containing proteins of interest were removed for MALDI-TOF mass spectrometry (Alberta Proteomics and MS facility at the University of Alberta).

## Immunostaining

BRGCs isolated from L3 staged at 40–44 h were dissected in 1X PBS buffer. Samples were then fixed with 1X PBS / 4% formaldehyde for 30 min and washed 3X with PBST (1X PBS with 0.1% Triton, Sigma Cat. # T9284) for 15 min per each round. Samples were then blocked for one hour in blocking solution (1X PBST with 5% goat serum) and washed 3X in PBST for 15 min each. Samples were next incubated with the primary antibody solution (anti-Flag from Cell signaling #8146 S with 1:800 dilution and anti-Hrs from DSHB #Hrs27-4 with 1:100 dilution, antibodies diluted in 1X PBST with 1% BSA) overnight at 4 °C. The following day, samples were washed 3X in BPST and incubated with the secondary antibody for one hour (Alexa Fluor 488 from Abcam #150113 with 1:2000 dilution and Alexa Fluor 555 from Abcam #150114 with 1:2000 dilution). Samples were then washed three additional times in

PBST and mounted in VECTASHIELD mounting medium (Cell signaling #4083). All images were taken with A Nikon C2si Confocal Microscope.

## LysoTracker Blue staining

For lysosome detection, BRGCs from 40 to 44 h L3 larvae were dissected in 1X PBS buffer. Samples were incubated for 30 min with LysoTracker Blue (Thermo Fisher Scientific, # L7525) a 1:10,000 dilution. Samples were incubated in the dark at room temperature, and on a shaker with slow agitation. Samples were then washed 3X with cold 1X PBS buffer (5 min each) and mounted in anti-Fade fluorescence mounting medium (Abcam, #ab104135). All images were taken with A Nikon C2si Confocal Microscope.

## Western blotting

We analyzed the RG, pull-down and hemolymph samples with standard Western blotting. The following antibodies were used to detect proteins on the blot. Flag-tagged proteins were detected with a monoclonal mouse anti-Flag antibody (Cell Signaling, #8146 S, 1:1000 dilution) followed by incubation with goat anti-mouse IgG H&L HRP secondary antibody (Abcam, #97023) at a ratio of 1:15,000. To detect Myc-tagged proteins, we used a monoclonal rabbit anti-Myc antibody (Cell Signaling, #2278 S, 1:1000 dilution), and blots were incubated with a goat anti-rabbit IgG H&L HRP secondary antibody (Abcam, #97051, 1:15,000 dilution). All protein bands were scanned and detected with the Bio-Rad ChemidocTM MP imaging system at the University of Alberta.

## Reporting summary

Further information on research design is available in the Nature Portfolio Reporting Summary linked to this article.

## Data availability

The source data for Figs. 1, 2, 3, 4, 5, 6, 7 and S6 are provided. Mass spectrometry proteomics raw data relating to Fig. 6a, Table 1, Supplementary Data 1 and Supplementary Data 2 have been deposited with the ProteomeXchange Consortium via the PRIDE partner repository with identifier PXD041626. Source data are provided with this paper.

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

## Acknowledgements

The authors thank the Bloomington *Drosophila* Stock Center at Indiana University and the Vienna *Drosophila* Resource Center for sending fly stocks. We thank the labs of Michael O'Connor and Gregory Emery for providing fly stocks. We would also like to thank Fanis Missirlis for sending stocks and providing insightful feedback on the manuscript. We further wish to thank also thank Dr. Arlene Oatway, Dr. Peng Li and Dr. Haoyang Yu for helping us to perform Transmission Electron Microscopy. We also extend our appreciation to Dr. Andrew Simmonds for his assistance in acquiring high-resolution images. Use of the Stanford Synchrotron Radiation Lightsource and the Linac Coherent Light Source, SLAC National Accelerator Laboratory, is supported by the US Department of Energy, Office of Science, Office of Basic Energy Sciences under Contract No. DE-AC02-76SF00515. The SSRL Structural Molecular Biology Program is supported by the DOE Office of Biological and Environmental Research, and by the National Institutes of Health, National Institute of General Medical Sciences (Grant P30GM133894). K.K.J. wishes to thank the CIHR (PS 169102) and NSERC (RGPIN-2018-04357) for supporting this work.

## Author contributions

S.S. co-designed experiments, conducted most experiments, and wrote the initial manuscript draft. S.M.W. and T.K. carried out the Synchrotron iron analysis at Stanford Synchrotron Radiation Lightsource. K.K.J. supervised and co-designed experiments, and co-wrote the manuscript with input from the other authors.

## Competing interests

The authors declare there are no competing interests.
