## [Peer Review File · Nature Communications]

Drosophila Evi5 is a Critical Regulator of Intracellular Iron Transport via Transferrin and Ferritin InteractionsREVIEWER COMMENTS

Reviewer #1 (Remarks to the Author):

This study follows from a previous clever RNAi-based screen in which the group used Fe-based PG autofluorescence to identify genes involved in heme synthesis and uptake. This approach has resulted in the identification and publication of several new genes and pathways, in this case *evi5*. Their extensive use of robust approaches indicate that *evi5* disrupts the normal process of Fe uptake and incorporation by PG cells by disrupting vesicular trafficking, in particular, in recirculation of endocytosed vesicles back to the membrane. Rather, these vesicles build up with unusual morphology and numbers, and contain what appear to be large condensates of ferritin iron complex intermediates. Co-culture with fat bodies, the use of GFP-tagged proteins and *in vivo* microinjections suggest that, under normal conditions, PG cells obtain their iron by the endocytosis of FB and hemolymph-carrying transferrin proteins, as well as possibly ferritin. The authors also do co-IPs to identify additional Evi5-interacting proteins with some strong candidates.

The study begins by demonstrating genetic interactions between *evi5*, *rab11* and *rab5*, which were quite convincing and biologically coherent. The consequences of disruption, in terms of vesicle and Fe particle buildups were very interesting, suggesting that lack of recycling eventually stalls further endocytosis and the progression of Fe-containing vesicles into compartments that can leverage the bound Fe. Clearly endocytosis wasn't the issue as FB-derived transferrin GFP levels rose significantly in PG cells, though apparently not giving up their cargos. Since release from transferrin appears to require low pH and lysosome trafficking, maybe failure to reach these organelles is where the transfer process is stalled. If true, then co-localization of transferrin and Fe with lysosome markers could be revealing.

I'm quite skeptical about their experiments in support of extracellular ferritin also being a major direct source of Fe. Previous studies have suggested that the major roles of large ferritin cages are as reservoirs and storage depots for Fe, thereby also helping to protect the host from excess Fe. I don't think any of the experiments carried out prove otherwise. There was limited if any observed ferritin-GFP endocytosis as modelled in Supp Fig 6B). The limited positive effects of injection experiments etc could be indirect, perhaps via transferrin. Unless further evidence is provided to support this, I think the authors need to dial this conclusion back, definitely removing the claim from the summary and discussions. Obtaining cellular Fe via ferritinophagy appears to be an intracellular event, not a means of obtaining extracellular Fe.

The pull-down, MS results were not really discussed or tested at length. The scores should be better described/annotated, for example providing full details on #s and identities of peptides detected. The small overlap in proteins identified using the different approaches, though potentially explainable, was worrisome. I'm particularly concerned about the significance of the Fer1HC pull-down, as it was very low

scoring and a potential indirect contamination. I wonder which peptide(s) were identified. This section was also weak due to minimal followup.

The authors end their discussion pointing to roles of Fe metabolism in multiple sclerosis, and how some of these aberrant developments resemble the consequences of Evi5 mutations. While I think this is a fair point to make, particularly to help emphasize impact, I suggest changing from calling MS “an Fe disease” to perhaps an Fe-exasperated disease.

In conclusion, I found the approaches used here to be quite robust, novel and complementary. Their analysis of Evi5 has illuminated new aspects of vesicle trafficking and Fe transport, which may be particularly important in tissues such as the PG where Fe metabolism is highly active and rate limiting. The data support roles in the uptake of Fe-bound transferrin and intracellular iron exchange. As such, while not ground breaking, it does provide a number of new insights. Additional points such as the AP-MS cofactor identifications added further depth, but it was more quantity than quality, as followup validations and function were very limited or lacking.

One significant control that appeared to be lacking but is important is the use of Westerns and immunofluorescence to show the robustness of PG RNAi/FRT deletions for Evi5, Rab5 and Rab11. Evi5 in particular should be shown to be expressed in PGs and how well the KD methods work. RGs can be readily dissected and collected in the presence of a little detergent and the use of BSA-coated tools and pipette tips.

The following are some additional minor issues:

p3. “evi-5 is critical for uptake of Fe”. Maybe not. Maybe for continued uptake of Fe. Seems its more important for keeping the vesicular cycle moving from unloading to return to the cell membrane.

I would like to see more follow up on IP'd proteins. As it stands, this is more of a weakness than help.

Though not essential, some further analysis of the accumulated vesicles and Fe condensates in evi5 mutants PGs might shed more light on what they are and why they are not releasing their Fe.

-In Fig. 5B, FB cells differed in size. Is this just stochastic, variation in sample selection or something going on?

p22 -The co-localization of Evi5 and N-terminal-truncated Tfns fused to GFP could be partly artifactual due to the manipulations of the genes.

Reviewer #2 (Remarks to the Author):

The manuscript by Soltani et al. explores the role of Evi5, a protein that activates at least one GTPase involved in endosome recycling. The model system under investigation is the prothoracic gland (PG) of *Drosophila melanogaster*.

Evi5 is one of the proteins that has emerged from a search (using RNAi lines) for factors contributing to iron homeostasis in the PG. The PG has a very high requirement for iron as it is the location of ecdysone production, which depends on several Fe-dependent enzymes, some of which are highly abundant.

The present manuscript reports on an impressive array of experiments to dissect Evi5's contribution to this important aspect.

Moreover, the human orthologue hEvi5 is an oncogene with involvement in vesicle trafficking, cell cycle regulation and cytokinesis.

This reviewer has no particular expertise in either insect biology and genetics or intricacies of iron trafficking, so they cannot comment on methodological details. Their comments should be seen in a more general context.

Overall, the nature, quality and quantity of the data presented are of a high standard and provide a number of novel insights. I was particularly intrigued by the ferritin-Evi5 interaction and the prospect that endocytosis of extracellular ferritin may be an alternative pathway for iron uptake, especially in the context of episodic peaks in iron requirements (for ecdysone synthesis) during different developmental stages.

A further intriguing aspect is the link to multiple sclerosis, where hallmarks of iron dyshomeostasis are also evident. Together with the fact that hEvi5 has emerged as a risk factor for MS from several GWAS studies, this seems very important and exciting.

Major issue

The paper would greatly benefit from an overview schematic that summarises the processes and major players discussed and how Evi5 fits into this bigger picture.

Minor issues

Page 2; Introduction. "Biological iron" should be summarized more accurately. As a minimum, there is a third kind of Fe protein (also non-heme), where catalytically active Fe is bound to amino acid sidechains. Typically, these are oxidoreductases including oxygenases. This important third kind of Fe-requiring enzymes should also be mentioned.

Page 2: Please spell out DMT1 or qualify what it is.

Page 3: does the short-chain dehydrogenase also require iron for function?

Page 3: For a general audience, it would perhaps be appropriate to define/explain "hemolymph" in half a sentence.

Page 5. I note that the citation regarding a causal link between MS and Fe dates from 1982. Whilst it is appropriate to cite this, it would be good to also include at least one more recent reference to make the point that this hypothesis is still "live".

Page 13: I was confused by the first few sentences of the Results section, as it appeared that Evi5 had been previously found via an RNAi screen reported in ref 19, but as far as I can tell, this part meant to indicate that a new RNAi screen was done for the present paper? I assume that the references were meant to refer to methodology rather than findings? It would be good to make this clearer.

Page 16: it was not obvious to me why particularly Rab5 and Rab11 were examined more closely (they ARE both mentioned in the introduction, but by the time a reader reaches page 16, this information may have evaporated - so a reminder may be helpful here).

Page 18: if allowed, it would be good to also spell out BRGC in the Results section, so the reader does not need to refer to the Methods part.

Page 18: "This increase was not due to a higher number in PG cells": please specify the number of what this refers to (perhaps vesicles?).

Page 20: “less Tsf1...when iron was scarce”. Is that as expected or the opposite? I seem to recall that in mammals, TF levels rise when Fe is scarce?

Bottom page 22/top of page 23: there is some repetition here that is probably not needed.

Page 22: “...Evi5 is linked to cellular iron biology” Why cellular and not systemic?

Page 24: bracket “in the form of equine spleen holo...” is in the wrong place – should be after the holo-, not the apo-.

Page 25 and elsewhere: please do not use the term “elemental iron”, as this suggests Fe(0) which clearly is not what you mean. Perhaps “non-protein-bound iron” is the least ambiguous description for this – or just call it FCA

Page 25: how was the total amount of administered Fe controlled, and was this the same for administration of holo-ferritin and FCA? If not the same, how does the difference impact the observations and conclusions from the experiments? I believe this warrants a mention in the results section.

Reviewer #3 (Remarks to the Author):

This is the manuscript that focuses on the role of endocytic traffic and Evi5 specifically, in regulating iron homeostasis in fly. Iron uptake and regulations is an important aspect for all organisms, thus, the manuscript is potentially interesting. However, its innovation is somewhat limited since there are numerous studies implicating Rab11 in regulating iron uptake and endocytic recycling of transferrin receptor in mammalian cells. The binding of Evi5 to ferritin is the most novel finding but the data demonstrating that is also pretty weak. Finally, manuscript also has some technical and quantification issues that should be fixed before study is published. Overall, in my opinion, the findings in this study is not novel enough for publication in journal of Nature Communications.

- 1) Figure 2F-G. The images and quantifications for 60 and 74 hour control also need to be shown. The presumptive iron depletion in Evi5-FRT mutants need to be compared to time-matched controls rather than earlier time points.

- 2) Figure 5. Knock out or constitutive activation of Rab11 had rather small effect on iron maintenance (unlike Evi5). That would suggest Rab11-independent function of Evi5. Can authors rescue Evi5 depletion with Evi5-Gap mutant?

- 3) Figure 4A-B. In this figure Hrs is used as EE marker, yet in the images (due to low resolution) it is impossible to distinguish EEs. What reviewers appear to quantify is overall fluorescence intensity that does not necessarily indicates number of EEs. Higher resolution images need to be shown and actual EEs need to be counted.

- 4) Figure 4C-D. Quantification and statistical analyses of vesicle size and number need to be performed.

- 5) Figure 5D needs to be quantified, followed by statistical analysis.

- 6) Table 1. I would have expected Rab11 to come down with Evi5, especially since it is considered to be GAP for Rab11.

- 7) Figure 6B. All immunoprecipitation experiments need to be run and blotted on the same gel. Displaying individual bands as cropped separate boxes is not acceptable.

We would like to thank the reviewers and the editor for carefully evaluating our manuscript, and for the valuable feedback we received. The comments have significantly contributed to improving overall clarity and robustness of our manuscript. Please find below a detailed point-by-point response addressing each of your comments (original comments are *blue and italicized*).

Reviewer #1:

*“The study begins by demonstrating genetic interactions between *evi5*, *rab11* and *rab5*, which were quite convincing and biologically coherent. The consequences of disruption, in terms of vesicle and Fe particle buildups were very interesting, suggesting that lack of recycling eventually stalls further endocytosis and the progression of Fe-containing vesicles into compartments that can leverage the bound Fe. Clearly endocytosis wasn’t the issue as FB-derived transferrin GFP levels rose significantly in PG cells, though apparently not giving up their cargos. Since release from transferrin appears to require low pH and lysosome trafficking, maybe failure to reach these organelles is where the transfer process is stalled. If true, then co-localization of transferrin and Fe with lysosome markers could be revealing.”*

Thanks for this really great suggestion! Indeed, since mutant Evi5 prothoracic glands (PG) suffer from impaired intracellular iron delivery, and the acidic environment in lysosomes is thought to promote the liberation of iron from transferrin, it is highly plausible that the acidification of endosomes, thus forming lysosomes, is impaired in these mutant cells, resulting in iron being stuck in these organelles. We attempted the following to address this. I) we stained lysosomes with LysoTracker Blue. We couldn’t use green or red LysoTracker dyes, since we were trying to detect transferrin-1-GFP in lysosomes and because the loss of Evi5 function resulted in red autofluorescence in the PG. This worked out very nicely, since mutant *PG>FLP;Evi5^{FRT}* animals displayed significantly weaker LysoTracker Blue staining compared to controls (see page 20 - highlighted in yellow- and Figures 4G and 4H), clearly showing either reduced levels of acidification. II) Subsequently, we attempted to assess the co-localization of Tsf1-GFP and LysoTracker Blue in our co-culture assay (page 23 and Figure S1E). Here, due to technical reasons, we were not able to demonstrate the possible co-localization of transferrin-1-GFP and lysosomes via LysoTracker Blue. Two factors were responsible for this. First, the lysosomal pH lies below the

working range of GFP, resulting in a hardly detectable GFP signal. To make matters worse, and as outlined above, Evi5 mutant cells had strongly reduced LysoTracker stains, making a co-localization with these tools impossible. We are showing the figure currently as Figure S1E and would appreciate some feedback as to whether we should delete the figure or keep it.

“I’m quite skeptical about their experiments in support of extracellular ferritin also being a major direct source of Fe. Previous studies have suggested that the major roles of large ferritin cages are as reservoirs and storage depots for Fe, thereby also helping to protect the host from excess Fe. I don’t think any of the experiments carried out prove otherwise. There was limited if any observed ferritin-GFP endocytosis as modelled in Supp Fig 6B). The limited positive effects of injection experiments etc could be indirect, perhaps via transferrin. Unless further evidence is provided to support this, I think the authors need to dial this conclusion back, definitely removing the claim from the summary and discussions. Obtaining cellular Fe via ferritinophagy appears to be an intracellular event, not a means of obtaining extracellular Fe.”

I agree that we do not provide direct evidence that ferritin is endocytosed by PG cells, but as I will outline below, I believe we have very strong indirect evidence that this must be occurring. However, I think it would be fair to say that our story is about Evi5 and transferrin, whereas the ferritin aspect came as a welcome bonus, which, in my opinion, suggests a larger role for Evi5 in iron metabolism as we originally thought. As such, finding solid evidence that Evi5 physically interacts with ferritin adds further credence to our story, since the big surprise was that Evi5, a multiple sclerosis-linked GWAS locus, regulates iron transport. Thanks to the additional link between Evi5 and ferritin, we are now designing a follow-up project to provide direct evidence for ferritin-mediated system iron transport – we can’t crack this nut in this story. In any case, our hypothesis about the utilization of ferritin as a source of iron that can be imported from the hemolymph is not really that far-fetched, and I will therefore try to address the reviewer’s concerns. It is true that historically, ferritin has been regarded as a sink for iron, and the general consensus was that iron was not retrievable once deposited into ferritin nanocages. However, with the discovery of the lysosomal degradation of ferritin (ferritinophagy), the paradigm changed and was replaced by the idea that ferritin can be disassembled via an autophagy-related process. Biologically this makes sense in situations where iron is scarce (why waste iron trapped in

ferritin?) or when cells require a large dose of iron in a short time (e.g., in preparation for a steroid hormone pulse due to the high abundance of cytochrome P450 enzymes, as we hypothesize in our manuscript).

In the insect iron field, ferritin has been considered an attractive and likely vehicle for systemic iron transport, for a couple of reasons. I) In insects, ferritin is predominantly secreted into the hemolymph, reaching approximately 1000-fold higher concentrations compared to ferritin in human blood (note that we were able to show that Fer1HCH-GFP is secreted into the hemolymph - see Figure S5B). II) midgut-specific disruption of ferritin expression resulted in reduced systemic iron levels, suggesting an alternative iron transport mechanism to transferrin. Even in humans, it has shown (citation #14, #15) that ferritin secretion utilizes two-non-canonical pathways, providing evidence ferritin export is – at least in part – a regulated process, rather than just the result of leakage from damaged cells.

Having said all this, direct evidence is lacking. However, our manuscript does provide additional indirect but arguably compelling evidence that point to ferritin as an iron transport vehicle, which is as follows:

1) In Figure 7A, we show a seemingly simple proof-of-principle experiment (i.e., ferritin injections rescue ferritin loss-of-function animals), but I believe it is very strong indirect evidence that extracellular ferritin is endocytosed by prothoracic gland cells. Specifically, the figure shows that PG-specific depletion of Fer1HCH via RNAi caused developmental arrest as second instar larvae (L2), and that injection of holo-ferritin (but not apo-ferritin) resulted in a substantial rescue, allowing the formation of adults. While it is tempting to interpret this result as a rescue of an iron deficiency in the PG, this very unlikely to be true. As the reviewer has outlined above, ferritin has been shown to act as a sink for iron because its main function is to detoxify cells by storing harmful excess iron. We believe that the following explanation is the most likely interpretation of the data in Fig. 7A. Apo-ferritin cannot enter cells, whereas holo-ferritin can (this could be either for physiological reasons, or simply due to the fact that the treatment to obtain apo-ferritin causes the ferritin molecule to be non-functional in this assay). Since holo-ferritin is rarely completely filled with iron atoms, we hypothesize that endocytosis of injected holo-ferritin has sufficient remaining capacity to store excess (toxic) iron, resulting in a rescue of the lethality. While this does not prove definitively that the PG can endocytose ferritin, I feel that this data is very difficult to explain

without invoking endocytosis. If one would argue that *PG>Fer1HCH-RNAi* causes iron deficiency in the prothoracic gland (which would be contradictory to its known function), then endocytosis of injected holo-ferritin would still be the most sensible explanation. Therefore, to come up with a scenario that does not involve endocytosis of ferritin, one would have to hypothesize the degradation of injected ferritin (via an unknown process, since the only described mechanism is lysosomal degradation, which requires intracellular ferritin) outside the prothoracic gland (but not in other tissues, since this would require also endocytosis), followed by the release of iron, which then somehow reaches the prothoracic gland. This seems convoluted and unlikely to me. By the way, iron feeding does not rescue *PG>Fer1HCH-RNAi* larvae, consistent with the idea that this is an iron detox phenotype.

2) When we injected *PG>FLP;Evi5^{FRT}* animals with various sources of iron (apo-, holo-, denatured holo-ferritin, and FAC, Figure 7B-D), holo-ferritin injections worked best across the board, with more dramatic reductions of developmental delays, red autofluorescence and ALAS expression. Remarkably, the iron content we injected via holo-ferritin is ~11,000 lower than the amount we injected via FAC, suggestive of a highly selective and targeted uptake mechanism for ferritin.

3) In Figure S5, we show rescue of the autofluorescence in *AGBE-RNAi* prothoracic glands due to the addition of hemolymph from larvae with other genotypes. I would like to point out two results here: i) The hemolymph for control (*w¹¹¹⁸*) animals rescues better than from *Fer1HCH-GFP* animals (compare S5A to S5D, either fourth row from top), consistent with the finding that the GFP moiety interferes with *Fer1HCH* function (see 4-ii). ii) The rescue is still substantial when we remove *Tsf-1* via RNAi, suggesting the rescue is caused by hemolymph ferritin.

4) To finish, we see four possibilities as to why we were unable to show direct uptake of *Fer1HCH-GFP* into the PG.

i) As mentioned in the discussion, the GFP : iron ratio in GFP-tagged ferritin (harbours up to 4,500 Fe^{3+}) is three orders of magnitude lower than that of *Tsf1-GFP* (carries one Fe^{3+})

ii) The GFP tag appears to interfere with ferritin function, since homozygotes are 100% lethal. This may disrupt certain aspects of ferritin function. For example, we know the ferritin-GFP can store iron and that it is excreted into the hemolymph, but it may affect endocytosis.

iii) The experiment was done cultured organs. If ferritin uptake is indeed the mechanism that provides large amounts of iron in preparation for a steroid hormone pulse (to ramp up heme co-factor production needed to equip cytochrome P450 enzymes), then we would not see endocytosed ferritin, because cultured ring glands do not produce ecdysone pulses (they lack systemic signals).

iv) Endocytosed ferritin is likely directly targeted for ferritinophagy, and ends up quickly in lysosomes, which as we have outlined earlier, interfere with the fluorescence of GFP due to denaturation.

“The pull-down, MS results were not really discussed or tested at length. The scores should be better described/annotated, for example providing full details on #s and identities of peptides detected. The small overlap in proteins identified using the different approaches, though potentially explainable, was worrisome. I’m particularly concerned about the significance of the Fer1HC pull-down, as it was very low scoring and a potential indirect contamination. I wonder which peptide(s) were identified. This section was also weak due to minimal followup.”

I agree that we have not taken full advantage of the MS pull-down experiments in the original submission, since we had limited our focus on the fact that Evi5 showed physical interaction with ferritin. However, I can’t say that I am concerned that the overlap is “small” or “worrisome”, or that the score for Fer1HCH is “very low”, and I hope that the following section will address these issues sufficiently.

We have now conducted a detailed term enrichment analysis of the MS data, and we have added a corresponding section to the manuscript (pages 24-25). Detailed MS data are now available in tables S3 and S4. One interesting observation was that the *in vivo* data was highly enriched for proteins known to physically interact with the Clathrin heavy chain (Chc), consistent with the idea that Evi5 is involved in receptor-mediated endocytosis. This enrichment was notably absent in the cell culture (*ex vivo*) MS data. The most-straightforward interpretation is that immortalized S2 cells have no need for receptor-mediated endocytosis since they lack the context of an organism and adjacent tissues. This is just one of the reasons as to why the *ex vivo* and *in vivo* data are different. A second major reason is that S2 cells originated from macrophages, and thus represent a single specialized cell type, whereas the *in vivo* data is based on all larval tissues – allowing for the detection of protein-protein interactions that do not occur in S2 cells (but in other cell types).

Even so, finding 16 overlapping proteins between the two cohorts (107 proteins and 75 proteins) is still highly significant (one would expect 0.6 proteins as the average overlap if one randomly chooses cohorts of 107 and 75 from a total of reported 13,824 *Drosophila* Uniprot proteins), which is virtually impossible to obtain randomly (p-value: $\sim 3 \times 10^{-92}$). We have added these stats to the figure. As a sidenote, the odds of finding ferritin heavy chain (= Fer1HCH) independently in both groups are 4.2×10^{-5} . I need to stress here that Fer1HCH was not present in either of the controls, indicating that Fer1HCH was not pulled down as a background contamination, and instead requires the presence of Evi5. For your reference, here are the peptides that were identified for Fer1HCH: "QEWTDGAAALSDALDLEIK," "LVEYLSMR", "TLKKMMDTNGELGEFLFDK," "GQLTEGVSDLINVPTVAK". All of them are a 100% match to Fer1HCH. With respect to the SEQUEST score, they are well within the normal range. A typical cutoff value for these MS data ranges from 1 to 3, but the scores for Fer1HCH are 8.17 and 7.58. Also note that Fer1HCH is ranked #53/107 and #38/75, which places it somewhere in the middle, score-wise. It may be that the Table 1 is a bit misleading as it only shows a subset of the detected proteins, positioning Fer1HCH lower than it actually is. Finally, we did validate the interaction between Evi5 and Fer1HCH using co-immunoprecipitation of tagged proteins (Figure 6B). For these reasons I feel confident that the interaction between Evi5 and Fer1HCH is real.

We added Rab11 to Table 1 to show that we did observe that it was pulled down in the experimental group. We added an asterisk to Rab11, because we also detected it in the control group (in the original figure, we Rab11 was not listed, prompting the question as to why we had not detected Rab11).

As a brief follow-up of the MS data, (highlighted in yellow on page 32) and illustrated in Figure S6B, we used PG-specific RNAi to knock down genes that may have roles in autophagy (and thus is the hypothesized ferritinophagy), including Clathrin heavy chain (Chc), Heat-shock protein (Hsc-70-4), as well as the well-known autophagy gene Atg8a. All of these knockdowns resulted in developmental defects, suggesting the importance of these genes in the prothoracic gland.

“The authors end their discussion pointing to roles of Fe metabolism in multiple sclerosis, and how some of these aberrant developments resemble the consequences of Evi5 mutations. While I

think this is a fair point to make, particularly to help emphasize impact, I suggest changing from calling MS “an Fe disease” to perhaps an Fe-exasperated disease.”

We have changed as follows: “Therefore, this study opens up the fascinating possibility that the reason Evi5 has been linked to MS lies in its role in cellular iron trafficking and that MS - at its core - is caused or exacerbated by a disruption of cellular iron transport.”

“One significant control that appeared to be lacking but is important is the use of Westerns and immunofluorescence to show the robustness of PG RNAi/FRT deletions for Evi5, Rab5 and Rab11. Evi5 in particular should be shown to be expressed in PGs and how well the KD methods work. RGs can be readily dissected and collected in the presence of a little detergent and the use of BSA-coated tools and pipette tips.”

I will start with a general comment. When dealing with knockdowns in *Drosophila*, we generally don't expect impressive reductions of the mRNA or the protein. This in turn begs the question of what constitutes a reasonable reduction in gene product. From experience in the fly field, researchers conducting qPCR of the mRNA targeted by RNAi often see a mere 50% reduction in mRNA levels, despite the presence of a strong phenotype (the best I have seen in my lab was 90%). Naively, this is what I would expect to see in a heterozygote... I recall a presentation at the US fly meeting where this was quantified across several hundred RNAi lines, where they found that the median effect of tested RNAi lines showed around 50% reduction. As such, it is uncommon that I ask my trainees to quantify the knockdown, because I expect that any reduction that is in this ballpark figure is unconvincing. Instead, we thrive to replicate the phenotype by at least one, non-overlapping RNAi construct, since our main concern is that the RNAi phenotype is triggered by an off-target. In this manuscript, we use three independent lines to validate the phenotypes of loss-of-Evi5 function. I would also like to stress that all the RNAi lines utilized in this study have undergone prior testing in other research studies and are readily available in stock centers. Detailed information about these lines can be found in table S1. Regarding the Evi5^{FRT} line (which produces a tagged version of Evi5, and can be excised via Flippase), we carried out a western blot using dissected ring glands as the reviewer requested (both from mutant (i.e., excised Evi5) and control (i.e., Evi5 not excised) animals. This showed significantly lower protein level in the mutant samples, indicating the depletion of Evi5 proteins after the induction of the Flippase enzyme in the

PG (highlighted in yellow on page 15 and shown in Figure S1C). Since we cannot separate the prothoracic gland from the other two glands in the ring gland, the Evi5 protein in the sample could originate from these glands.

As for the Rab11 and Rab5 RNAi lines, we could not carry out a similar Western blot analysis due to the unavailability of commercial antibodies for these proteins in *Drosophila*. Nevertheless, we have confidence in their functionality based on the observed developmental arrest or delay phenotypes and red autofluorescent phenotype when these lines were. The autofluorescence phenotype is overall rare and very specific, and as such a sign of specificity. Additionally, we used *PG>W¹¹¹⁸* animals as controls in all experiments to ensure that the expression of an excess amount of GAL4 protein did not result in any unexpected phenotypes.

“evi-5 is critical for uptake of Fe”. Maybe not. Maybe for continued uptake of Fe. Seems its more important for keeping the vesicular cycle moving from unloading to return to the cell membrane.”

Good point. We have revised the statement to "critical for maintaining iron uptake from the hemolymph" to ensure that our emphasis is not only on the uptake process.

“In Fig. 5B, FB cells differed in size. Is this just stochastic, variation in sample selection or something going on?”

There is indeed quite a bit of variation. Even though we dissected these samples using the same absolute times since egg laying, it is not uncommon that the population is not synchronized anymore by the time they reach later larval stages. When we need to ensure tighter developmental synchronization, there is a way for us to do it (which quite labor-intensive), but in this case we simply wanted a qualitative assessment of GFP absence/presence, and thus did not opt for synchronizing the population.

“p22 -The co-localization of Evi5 and N-terminal-truncated Tfns fused to GFP could be partly artifactual due to the manipulations of the genes.”

Indeed, this is a somewhat artificial setup, which is why we decided to present these results only in a supplemental figure. However, testing the co-localization of Evi5 with Tsf proteins is challenging, mainly because Tsf proteins are secreted from cells, so without the truncation, the co-localization would be short-lived and thus difficult to detect (example for Tsf1 is shown in Figure S2C).

Reviewer #2:

“The paper would greatly benefit from an overview schematic that summarises the processes and major players discussed and how Evi5 fits into this bigger picture.”

Done! Please see Figure S6A.

“Introduction. “Biological iron” should be summarized more accurately. As a minimum, there is a third kind of Fe protein (also non-heme), where catalytically active Fe is bound to amino acid sidechains. Typically, these are oxidoreductases including oxygenases. This important third kind of Fe-requiring enzymes should also be mentioned.”

A fair point! We have rephrased the section as follows. “Biological iron is most commonly found in two types of protein cofactors, heme and iron-sulfur clusters (Fe-S), and act in various electron transfer reactions. Less common are proteins that bind the metal directly, such as mononuclear (non-heme) iron oxygenases or dinuclear iron centers present in some ferroxidases.

“Page 2: Please spell out DMT1 or qualify what it is.”

We changed DMT1 to “Divalent Metal Transporter 1 (DMT1)” on page 2.

“Page 3: does the short-chain dehydrogenase also require iron for function? “

No. The short-chain dehydrogenase in the ecdysone synthesis pathway is named “Shroud” and it does require iron/heme cofactors but instead relies on NADPH as a cofactor for its catalytic activity. For improved clarity, the sentence now reads “The Halloween genes encode six P450 enzymes (which bind heme), a single short-chain dehydrogenase/reductase (does not require iron) and a Rieske electron oxygenase that harbors an Fe-S cluster.”

“Page 3: For a general audience, it would perhaps be appropriate to define/explain “hemolymph” in half a sentence.”

Done. On page 3 it now states "In this study we show that the Ecotropic viral integration site 5 (Evi5) is a hitherto unidentified regulator of vesicular iron transport in *Drosophila*, and critical for maintaining iron uptake from the hemolymph, the equivalent of blood in insects. "

“Page 5. I note that the citation regarding a causal link between MS and Fe dates from 1982. Whilst it is appropriate to cite this, it would be good to also include at least one more recent reference to make the point that this hypothesis is still “live”.”

Fixed. We have added recent references (references: #55 and #56) to support the ongoing relevance of the studies linking MS and iron.

“Page 13: I was confused by the first few sentences of the Results section, as it appeared that Evi5 had been previously found via an RNAi screen reported in ref 19, but as far as I can tell, this part meant to indicate that a new RNAi screen was done for the present paper? I assume that the references were meant to refer to methodology rather than findings? It would be good to make this clearer.”

We have improved this section. There were indeed two screens, only one of which was published, and focused on identifying genes with important functions in the *Drosophila* prothoracic gland. In a subsequent, unpublished screen, we searched a subset of ~800 lines for either red autofluorescence in or enlarged growth of the ring gland. The beginning now reads: “To identify new components of iron metabolism and iron transport, we conducted an RNAi screen in the *Drosophila* PG, a tissue with unusually high iron demands. Specifically, we selected 803 RNAi

lines that showed larval lethality in a prior PG-specific screen⁶⁵, and searched for the presence of red autofluorescing PGs, which is indicative of heme precursor accumulation due to incomplete heme synthesis²¹”.

“Page 16: it was not obvious to me why particularly Rab5 and Rab11 were examined more closely (they ARE both mentioned in the introduction, but by the time a reader reaches page 16, this information may have evaporated - so a reminder may be helpful here).”

Good point! To address this, we have expanded the section as follows: "Rab proteins are small GTPases that are critical for regulating membrane traffic, including endosome recycling, exocytosis, and Golgi/endosome transport processes. In vertebrates, Rab5 and Rab11 are critical for cellular cargo delivery via endocytosis and exocytosis, including iron^{76,77}. Both Rab5 and Rab11 have been linked to the endocytosis of transferrin, since Rab5 co-localized with the transferrin receptor⁷⁸, whereas Rab11 was required for cellular transferrin transport back to the cell surface⁷⁹. Evi5 functions as a GTPase-activating protein GAP for Rab11 and binds directly to the GTP-bound form of Rab11 to coordinate vesicular trafficking⁴²."

“Page 18: if allowed, it would be good to also spell out BRGC in the Results section, so the reader does not need to refer to the Methods part.”

Not sure if allowed, but I definitely agree, and we have now added “brain-ring gland complexes (BRGCs)” to the results section for better clarity.

“Page 18: “This increase was not due to a higher number in PG cells”: please specify the number of what this refers to (perhaps vesicles?).”

The mentioned increase refers to the Hrs signal density. To improve clarity, we have revised the sentence as follows: "This increase of Hrs signal density was not due to a higher number in PG cells, which were comparable between controls and Evi5-loss-of-function PGs (Fig. 4C)."

“Page 20: “less Tsf1...when iron was scarce”. Is that as expected or the opposite? I seem to recall that in mammals, TF levels rise when Fe is scarce?”

When iron levels drop, iron regulatory protein 1 undergoes a conformational change and becomes RNA-binding, and known bound transcripts include the transferrin receptor, DMT1 and the ferritin genes. Transferrin (TF) itself is not regulated by this mechanism, but possibly by other pathways. So, I am not quite sure if there is a consensus on how transferrin expression is regulated or in which direction it goes. In any case, the statement was based on the co-culture assay, where we observed lower GFP signals (= endocytosed Tsf1-GFP) in PG cells of the *PG>FLP;Evi5^{FRT}* that were co-cultured with Tsf1-GFP fat body tissues in the medium with the iron chelator BPS. So, in this statement we are simply saying the less transferrin arrives in the PG, it is not a statement about whether or not there is upregulation of Tsf1 in response to iron deprivation. Note that an upregulation of Tsf1 is therefore compensatory only and does not mean that more iron is transported (in fact that would be paradoxical).

Having said this, our findings do suggest, as the reviewer suspects, that Tsf1 expression did indeed increase in fat bodies (presumably to snatch up that rare iron atom) when they were cultured in an iron-depleted medium (Figure 5B). Nonetheless, this likely means that the majority of the Tsf1 proteins were in the apo-form due to the lack of Fe²⁺ in the medium, resulting in less transferrin being released and causing a reduction of transferrin in the co-cultured prothoracic gland.

“Bottom page 22/top of page 23: there is some repetition here that is probably not needed”

Here, we are unfortunately not quite sure what the reviewer had in mind and would need clarification on this point to address this.

“Page 22: “...Evi5 is linked to cellular iron biology” Why cellular and not systemic?”

The use of "cellular" rather than "systemic" usually refers to different concepts in the iron field, where systemic iron homeostasis is regulated by a hormone (hepcidin) and cellular iron homeostasis by iron regulatory proteins (e.g., IRP1). Since Evi5 acts within cells by regulating vesicular transport of iron, we think of this protein as a component of cellular iron biology, rather than systemic iron biology. The particular section you are referring to was based on the

identification of Fer1HCH in the IP-MS experiments (which occurs with intracellular ferritin, not the secreted form of ferritin). For better clarity, we changed the statement to: "The identification of Fer1HCH in two independent IP-MS experiments raised the possibility that Evi5 has roles in iron metabolism independent of vesicular transport of transferrins."

"Page 24: bracket "in the form of equine spleen holo..." is in the wrong place – should be after the holo-, not the apo-."

Thank you! We have corrected the typo on page 27. It now states: "To accomplish this, we injected control buffer (PBS), as well as holo- (iron-loaded) and apo- (iron-free) ferritin (in the form of equine spleen ferritin) into second instar larvae (L2)."

"Page 25 and elsewhere: please do not use the term "elemental iron", as this suggests Fe(0) which clearly is not what you mean. Perhaps "non-protein-bound iron" is the least ambiguous description for this – or just call it FCA"

Thank you for pointing this out! We have removed the reference to elemental iron and replaced it with "non-protein-bound iron (in the form of FAC)."

Reviewer #3:

"1) Figure 2F-G. The images and quantifications for 60 and 74 hour control also need to be shown. The presumptive iron depletion in Evi5-FRT mutants need to be compared to timematched controls rather than earlier time points."

We understand that a comparison would be nice here, however, at the 60- and 74-hour time points (relative to the L2/L3 molt), the control larvae have already formed pupae. In *Drosophila*, 3rd instar larvae undergo puparium formations approximately 48 hours after the L2/L3 molt. As a result, time-matched comparisons between control and *PG>FLP; Evi5^{FRT}* animals at these time points are not feasible, as the controls have entered the next developmental stage. Puparium formation comes with massive developmental changes, tissues are histolyzed or remodeled,

including the brain, the ring gland the fat body, the gut, and so on. Our intention was to continue observing the *PG>FLP; Evi5^{FRT}* animals for a couple of additional time points to assess iron levels in the ring gland and the brain. We couldn't go beyond these time points since Synchrotron time is extremely limited and expensive. In any case, we have changed the label of Figure 2F-G to "animals pupariated" to better convey the fact that the wild type controls have entered a different developmental stage that cannot be properly compared to the Evi5 loss of function animals.

"2) Figure 5. Knock out or constitutive activation of Rab11 had rather small effect on iron maintenance (unlike Evi5). That would suggest Rab11-independed function of Evi5. Can authors rescue Evi5 depletion with Evi5-Gap mutant?"

This is certainly a possibility, and we went to great lengths to test this. While there was a previously published Evi5-GAP mutant transgenic line available (citation #44), it would not have worked in our assays, since that particular line produces an Evi5 variant fused to mCherry, which would have interfered with the rescue of the red autofluorescence due to protoporphyrin accumulation (i.e., we would not have been able to assess a reduction in protoporphyrin accumulation due to mCherry masking the reduction of red autofluorescence). To address the reviewer's question, we needed to generate two new transgenic lines for Evi5, which were equivalent to the existing lines but instead produced Evi5 proteins coupled to mVenus. One variant was the wild type Evi5 cDNA, whereas the second construct harboured a point mutation resulting in an Arg160 to Ala substitution (Evi5^{R160A}), which is a catalytically inactive form of Evi5, as described in the methods section and Laflamme et al.'s 2012 study (citation #44).

The next challenge was that we needed to combine the new transgenic lines with the *Evi5^{FRT}* line. We considered using the Evi5 RNAi line (which would have been simpler), but we were worried that the Evi5 RNAi would target the transgenically produced Evi5 mRNAs as well, potentially weakening the phenotype. As such, using the Evi5^{FRT} was the most challenging option, but at the same time the cleanest solution. This involved a series of complicated genetic crosses with *UAS-Flippase*, *UAS-Evi5^{WT}* or *UAS-Evi5^{R160A}*, and *Phm-GAL4* drivers to express Evi5^{WT} or Evi5^{R160A} cDNAs while depleting endogenous Evi5 simultaneously. The results, presented in the new Figure 3F and described on Page 18, revealed that expressing wild-type Evi5 cDNA rescued the red autofluorescence phenotype of *PG>FLP;Evi5^{FRT}* animals, whereas the catalytically inactive form

failed to rescue this phenotype (Figure 3F). This demonstrated that the GAP domain of Evi5's was essential for this process.

It is noteworthy that Rab11 is involved in various vesicular trafficking processes, and its function is not limited to vesicular trafficking linked to iron uptake and metabolism. We further tested all known Rab11 GEF effector proteins (Figure 3H) and our results demonstrated that none of the PG-specific loss of function experiments of the known Rab11-GEF proteins caused red autofluorescence. Additionally, we examined a constitutively active form of Rab11 (Rab11^{CA}) that mimics the Evi5 loss of function phenotype. *PG>Rab11^{CA}* animals showed only a minor developmental delay with no red autofluorescence in PG cells. Taken together, these results strongly support the specificity of Evi5's role in vesicular trafficking of iron.

“3) Figure 4A-B. In this figure Hrs is used as EE marker, yet in the images (due to low resolution) it is impossible to distinguish EEs. What reviewers appear to quantify is overall fluorescence intensity that does not necessarily indicates number of EEs. Higher resolution images need to be shown and actual EEs need to be counted.”

Thank you for pointing this out! We acknowledge that while overall Hrs fluorescence may not precisely correspond to the number of early endosomes (EE) in PG cells, it does reliably correlate with vesicle abundance. To improve clarity, we have revised the text to read: "BRGCs from *PG>FLP;Evi5^{FRT}* and *PG>Evi5^{IR1}* larvae showed 3-4 times higher Hrs levels than controls (79.4 and 63.5 vs. 22.5 signal density), indicating EE accumulation (Fig. 4A-C)," as indicated on page 20. Furthermore, we have updated the labels for the quantified Hrs signals in Figure 4C to "signal density/RG" to emphasize that we measured the Alexa Fluor 488 signal in the RG samples of the tested animals. To directly address the reviewer's request to obtain higher resolution images, we tried various methods and added new images, as shown in Figure 4B. We then sought assistance from Dr. Andrew Simmonds (chair of the Department of Cell Biology at the University of Alberta), who works on peroxisomes and is very experienced in a range of the latest microscopy techniques. After discussions with Dr. Simmonds regarding EE detection, we decided to use the Leica Thunder widefield microscope to capture 3D images with minimal photobleaching and phototoxicity, followed by a deconvolution analysis to enhance signal resolution. Despite acquiring images with

~100 to 150 Z-slices, the assembled images did not exhibit the expected improvement in resolution. Here are two examples of captured images:

Therefore, we decided to use our confocal microscope instead, and made efforts to capture high-resolution images by adjusting the Z-stack parameters for RG samples. Initially, we analyzed the images using the Huygens Confocal Deconvolution Software to determine the optimal Nyquist rate and PSF parameters. Subsequently, we acquired new images with the identified parameters and conducted a deconvolution analysis using the Huygens confocal deconvolution software. Results are now shown in Figure 4B.

“4) Figure 4C-D. Quantification and statistical analyses of vesicle size and number need to be performed.”

We have now quantified both the size and number of vesicles in the TEM images. The corresponding results are presented in the new Figure 4F, and explanations can be found on page 20.

“5) Figure 5D needs to be quantified, followed by statistical analysis.”

Done! In Figure 5D, we have quantified the GFP signal in the RG and CNS of all samples, as illustrated in bar graphs and described on page 22. Following the quantification, a student’s t-test was performed between the control and experimental samples to assess statistical significance.

“6) Table 1. I would have expected Rab11 to come down with Evi5, especially since it is considered to be GAP for Rab11.”

We indeed we were able to pull down Rab11 with Evi5 in both in vivo and ex vivo tests. However, Rab11 was unfortunately also identified in one of the control groups, and as we mentioned in the text, "we removed all proteins from the final list if they appeared in the control and experimental samples" (page 24), Therefore, we had originally excluded Rab11 from the final hits. However, since you raise a fair point, we have now added Rab11 to Table 1, and use an asterisk to indicate that it was also detected in one of the controls.

“7) Figure 6B. All immunoprecipitation experiments need to be run and blotted on the same gel. Displaying individual bands as cropped separate boxes is not acceptable.”

Thank you for your comment regarding Figure 6B. Please rest assured that all immunoprecipitation experiments were indeed run and blotted on the same gel. We have included all uncropped blots in Figure S7.

REVIEWERS' COMMENTS

Reviewer #1 (Remarks to the Author):

The authors have responded to most of the reviewer's criticisms and requests, making the manuscript quite a bit stronger. That said, there are still lingering concerns. For example, many answers were provided in the form of long winded explanations rather than clear cut experiments. An example is the refusal to test RNAi effectiveness via Westerns, or Evi5 expression levels in in both WT and mutant PGs. The response that the PI does not ask trainees to test this because the average RNAi line only knocks out expression by ~50% is not a satisfying one. Most lines that reduce expression by 50% or less are actually not useful, which is not surprising in that most genes function fine as heterozygotes. And even though 3 lines were used, it would be helpful to show the correlation between RNAi efficiency and phenotype strength. The author's explanation makes me feel that this was tried but the results failed to work or to correlate with the phenotype. There were numerous other instances where lists of arguments were made rather than finding approaches that can directly validate things. For example, it was stated that GFP fluorescence could not be assessed in lysosome due to the low pH - how about using anti-GFP Abs instead? I was also not happy about Rab11 being included in the AP-MS tables, since it also showed up with the negative control (note that Rab11 was a key expected positive control). One of the arguments for lack of uniformity in the identified protein sets was due to the use of S2 cells, which come from a macrophage origin. Shouldn't a more relevant cell type be used then?

I am also still somewhat skeptical about the interaction between Evi5 and ferritin. On this point, I noticed that the authors did not comment on Reviewer 3's statement that the manuscripts: "innovation is somewhat limited since there are numerous studies implicating Rab11 in regulating iron uptake and endocytic recycling of transferrin receptor in mammalian cells. The binding of Evi5 to ferritin is the most novel finding but the data demonstrating that is also pretty weak. Overall, in my opinion, the findings in this study is not novel enough for publication in journal of Nature Communications." The author did touch on these points in their response to my critiques with a sprinkling of new data and arguments, and the case does make a lot of sense, but again, most of the actual evidence is still indirect or weak. Given that this may be the most novel and important contribution of this study, it seems it should be addressed more directly. Thus, unless some of these points are better addressed, I still see myself sitting on the fence re acceptance.

Reviewer #2 (Remarks to the Author):

The authors have thoroughly addressed my concerns.

As for the very minor comment regarding repetition, this had referred to the explanations regarding ferritin (with similar but less detailed information on page 3), but upon reviewing this again, a change is not necessary.

Reviewer #3 (Remarks to the Author):

Authors have addressed most of the concerns/suggestions raised by the reviewers. In my opinion, the manuscript is now ready for publication.

Blue italics: comments from the reviewers

Reviewer #1 (Remarks to the Author):

The authors have responded to most of the reviewer's criticisms and requests, making the manuscript quite a bit stronger. That said, there are still lingering concerns. For example, many answers were provided in the form of long winded explanations rather than clear cut experiments. An example is the refusal to test RNAi effectiveness via Westerns, or Evi5 expression levels in in both WT and mutant PGs. The response that the PI does not ask trainees to test this because the average RNAi line only knocks out expression by ~50% is not a satisfying one. Most lines that reduce expression by 50% or less are actually not useful, which is not surprising in that most genes function fine as heterozygotes. And even though 3 lines were used, it would be helpful to show the correlation between RNAi efficiency and phenotype strength. The author's explanation makes me feel that this was tried but the results failed to work or to correlate with the phenotype.

Let me start by saying that we are not trying to hide anything - we usually don't incorporate the quantification of RNAi effectiveness (other than on a phenotypic level) into our workflow. One point I would like to correct first: Many Drosophila RNAi lines show a specific phenotype (i.e., validated by the mutant phenotype) despite showing only 50% transcript level reduction on qPCR data. Which is puzzling, precisely because one would expect such a mediocre reduction to not be associated with a mutant phenotype. Given the error bars associated with qPCR quantification, it is perhaps understandable that I find qPCR results showing moderate reduction in an RNAi line hard to interpret.

We do, however, carry out mRNA quantification of RNAi lines, IF there is NO observable phenotype, in order to answer the question whether the RNAi line is working in the first place. If we do however, observe a phenotype, we know that the RNAi is working per se, and what needs to be done now is to examine whether the observed phenotype is reproducible and not caused by an off target. Off-targets are our primary concern when using RNAi, and we typically spend a considerable amount of effort in validating the specificity of the RNAi phenotypes. This is done by independent lines and independent methodologies, where we use a non-overlapping RNAi construct and an RNAi-independent approach (here, an FLP/FRT-excisable Evi5 allele). With three independent approaches in this paper showing the same, exceedingly rare and specific phenotype (red autofluorescence due to protoporphyrin accumulation) in the prothoracic gland (PG), the specificity of this loss-of-function phenotype is beyond reasonable doubt.

The second major reason as to why we do not use quantification of mRNA or protein in this case is the fact that the PG cannot be separated from the other two tissues that the gland is fused to (PG + CA + CC = ring gland, see Figure S1B). As such, any ring gland samples one would use to assess Evi5 transcript or protein levels would comprise

tissue where the RNAi was expressed (PG) and tissue where Evi5 expression would be wild type (since we use a PG-specific driver to express Gal4 (for RNAi) and FLP (for the Evi5^{FRT} allele).

I do want to point out that we did try to accommodate this reviewer's suggestion and we did quantify the reduction of Evi5 protein in the ring glands isolated from the Evi5^{FRT} line (Figure S1C). As outlined above, the prothoracic gland cannot be separated from the adjacent glands, but Figure S1C clearly shows a reduction of Evi5 protein, whereas the control (tubulin) has the same level in both samples. Whether the remaining band originates from residual protein in the prothoracic gland or from the adjacent tissues cannot be determined in this Western.

There were numerous other instances where lists of arguments were made rather than finding approaches that can directly validate things. For example, it was stated that GFP fluorescence could not be assessed in lysosome due to the low pH - how about using anti-GFP Abs instead? I was also not happy about Rab11 being included in the AP-MS tables, since it also showed up with the negative control (note that Rab11 was a key expected positive control). One of the arguments for lack of uniformity in the identified protein sets was due to the use of S2 cells, which come from a macrophage origin. Shouldn't a more relevant cell type be used then?

i) To show co-localization with lysotracker signals, we need the GFP signal - a antibody stain would be incompatible with the lysotracker stain.

ii) With respect to using another cell type: Sure one can do this, but all cell lines are immortalized and therefore genetically altered compared to the founder cell population. Also, cell culture samples only represent a specific cell type. As such, I would not be optimistic about finding a higher degree of overlap between cell culture and animal samples if one were to switch from S2 cells to some other Drosophila cells. Comparing whole animal samples to any given specific cell type will always result in substantial differences. Despite all this, the overlap is still highly significant, and given the difference in the underlying nature of the samples, completely in line with my expectations.

iii) We included Rab11 in the table to satisfy the concerns of another reviewer.

I am also still somewhat skeptical about the interaction between Evi5 and ferritin. On this point, I noticed that the authors did not comment on Reviewer 3's statement that the manuscripts: "innovation is somewhat limited since there are numerous studies implicating Rab11 in regulating iron uptake and endocytic recycling of transferrin receptor in mammalian cells. The binding of Evi5 to ferritin is the most novel finding but the data demonstrating that is also pretty weak. Overall, in my opinion, the findings in this study is not novel enough for publication in journal of Nature Communications." The author did touch on these points in their response to my critiques with a sprinkling of new data and arguments, and the case does make a lot of sense, but again, most of the

Response to reviewers (revision NCOMMS-23-11462A)

actual evidence is still indirect or weak. Given that this may be the most novel and important contribution of this study, it seems it should be addressed more directly. Thus, unless some of these points are better addressed, I still see myself sitting on the fence re acceptance.

Looking at the response of reviewer #3, he/she seems to be satisfied with the changes to the manuscript. As I outlined in my previous response, I think we have done what can be reasonably expected for this initial characterization of the link between Evi5 and iron transport, and demonstrated protein interaction between Evi5 and Fer1HCH (not technically the same as ferritin) by independent procedures - I do not consider this aspect weak at all. However, to understand the nature of this interaction and dissect the underlying biology requires a dedicated follow-up project on this subject.

I disagree with your assessment of the importance of our findings. This is highly subjective, of course, and hard to address directly. In any case, has anybody ever linked Evi5, a human GWAS hotspot for MS, to iron biology? We firmly established that Evi5 plays a crucial role in transferrin-mediated uptake of iron, which also demonstrates - for the first time - that insect transferrins indeed transport iron systemically (recall that the transferrin receptor is famously missing in insects and other non-vertebrates - as such it is was one of the big unsolved questions - do insect transferrins transport iron at all?). Importantly, the finding that Evi5 interacts with Fer1HCH adds another dimension to the Evi5-transferrin results and we decided to present these data to strengthen the link between Evi5 and iron. We can't address everything in this first report, but we believe that the data strongly supports of a link between Evi5 and iron biology. We will continue to work on the ferritin aspect, and we will explore whether ferritinophagy exists in insects, and how Evi5 relates to all this. But that will be a different publication...

Thank you!

Reviewer #2 (Remarks to the Author):

The authors have thoroughly addressed my concerns. As for the very minor comment regarding repetition, this had referred to the explanations regarding ferritin (with similar but less detailed information on page 3), but upon reviewing this again, a change is not necessary.

Great! Thank you!

Reviewer #3 (Remarks to the Author):

Response to reviewers (revision NCOMMS-23-11462A)

*Authors have addressed most of the concerns/suggestions raised by the reviewers.
In my opinion, the manuscript is now ready for publication.*

Thank you very much!